# Marine Ice Sheet Instability and Ice Shelf Buttressing Influenced Deglaciation of the Minch Ice Stream, Northwest Scotland

Niall Gandy[1], Lauren J. Gregoire[1], Jeremy C. Ely[2], Christopher D. Clark[2], David M. Hodgson[1], Victoria Lee[3], Tom Bradwell[4], Ruza F. Ivanovic[1]

[1]School of Earth and Environment, The University of Leeds, Leeds, LS2 9JT, United Kingdom
[2]Department of Geography, The University of Sheffield, Sheffield, S10 2TN, United Kingdom
[3]School of Geographical Sciences, The University of Bristol, Bristol, BS8 1SS, United Kingdom
[4]Biological and Environmental Sciences, The University of Stirling, Stirling, FK9 4LA, United Kingdom

*Correspondence to*: Niall Gandy (eeng@leeds.ac.uk)

**Abstract.** Uncertainties in future sea level projections are dominated by our limited understanding of the dynamical processes that control instabilities of marine ice sheets. The last deglaciation of the British-Irish Ice Sheet offers a valuable example to examine these processes. The Minch Ice Stream, which drained a large proportion of ice from the northwest sector of the British-Irish Ice Sheet during the last deglaciation, is constrained with abundant empirical data which can be used to inform, validate and analyse numerical ice sheet simulations. We use BISICLES, a higher-order ice sheet model, to examine the dynamical processes that controlled the retreat of the Minch Ice Stream. We perform simplified experiments of the retreat of this ice stream under an idealised climate forcing to isolate the effect of marine ice sheet processes, simulating retreat from the continental shelf under constant "warm" surface mass balance and sub-ice shelf melt. The model simulates a slowdown of retreat as the ice stream becomes laterally confined at the mouth of the Minch Strait between mainland Scotland and the Isle of Lewis, resulting in a similar marine setting to many large tidewater glaciers in Greenland and Antarctica. At this stage of the simulation, the presence of an ice shelf becomes a more important control on grounded ice volume, providing buttressing to upstream ice. Subsequently, the presence of a reverse slope inside the Minch Strait produces an acceleration in retreat, leading to a 'collapsed' state, even when the climate returns to the initial 'cold' conditions. Our simulations demonstrate the importance of the Marine Ice Sheet Instability and ice shelf buttressing during the deglaciation of parts of the British-Irish Ice Sheet. We conclude that geological data could be applied to further constrain these processes in ice sheet models used for projecting the future of contemporary ice sheets.

## 1 Introduction

Attempts to model the future evolution of the West Antarctic Ice Sheet reveal large uncertainty in the extent of future mass loss (Feldmann and Levermann, 2015; Ritz et al., 2015). This is partly because many contemporary Antarctic Ice Streams are marine based (Jenkins et al., 2010; Joughin et al., 2014; Ross et al., 2012), and are therefore vulnerable to Marine Ice Sheet Instability (MISI). Schoof (2007) demonstrated that no stable grounding line position is possible in areas of reversed bed slope.

Consequently, any change in ice thickness at the grounding line can cause an irreversible grounding line migration with no change in external forcing. However, it has been shown that simulations of grounding line migration require not only consideration of bed topography, but also ice shelf buttressing (Gudmundsson, 2013), which can stabilise grounding lines on reverse sloping beds. The previous generation of ice sheet models do not accurately simulate the position of the grounding line, due to the use of the Shallow Ice Approximation (van der Veen, 2013), although significant improvements have been made using subgrid parameterization at the Grounding Line (Feldmann et al., 2014). Higher-order models have more success in accurately simulating the grounding line (Favier et al., 2014; Pattyn et al., 2012), but are still sensitive to model resolution (Cornford et al., 2016) and the representation of basal sliding processes (Gladstone et al., 2017; Nias et al., 2016; Tsai et al., 2015).

It is essential for improved future predictions of ice sheet change to better understand the dynamics of marine ice sheets over millennial timescales. A numerical simulation of the palaeo Marguerite Bay Ice Stream since the Last Glacial Maximum shows that the episodic retreat was controlled by a combination of bed topography, ice stream width, and upstream response, and that these controls are crucial to understanding centennial ice sheet evolution (Jamieson et al., 2012). A valuable case to examine these processes is the last deglaciation of the British-Irish Ice Sheet (BIIS), which had a number of marine-grounded sectors (Clark et al., 2012). While contemporary ice sheets offer a decadal scale observational record, the palaeo record of the BIIS provides detailed proxy observations of ice sheet retreat over millennia. The behaviour of the BIIS has been studied for over a century, resulting in much information on flow patterns and margin positions against which ice-sheet models can be compared (Clark et al., 2018). Information on the timing and pace of retreat has been considerably enhanced through the recent work of the BRITICE-CHRONO project, a multi-organization consortium, which has collected data to better constrain the timing of retreat of the BIIS, particularly in marine sectors. The data-rich environment that the empirical record now holds makes it an attractive test bed for numerical ice sheet modelling experiments (e.g. Boulton et al., 2003; Boulton and Hagdorn, 2006; Hubbard et al., 2009; Patton et al., 2016, 2017). The modelling investigations of Boulton et al. (2003) and Hubbard et al. (2009) specifically highlighted the importance of ice stream dynamics in the evolution of the ice sheet. However, simulations of marine ice sheets, like the West Antarctic Ice Sheet or the BIIS, ideally require models that are able to simulate grounding line migration (Pattyn et al., 2012). The BISICLES ice sheet model was developed to efficiently and accurately model marine ice sheets (Cornford et al., 2013), allowing for new simulations of the BIIS which explore marine influence on the ice sheet.

We investigate the marine influence on one very well constrained ice stream of the BIIS, the Minch Ice Stream (MnIS), using the BISICLES ice sheet model. Here, we perform and analyse numerical modelling simulations to test two hypotheses of MnIS retreat; (1) that the ice stream experienced MISI, and (2) that an ice shelf had an influential buttressing effect on the pace of retreat.

## 2 The Minch Ice Stream

The Minch Ice Stream (MnIS) flowed northward from the northwest Scottish Highlands through the Minch Strait (Figure 1). It reached its maximum extent, the edge of the continental shelf, at ~27 ka BP (Bradwell et al., 2008), which we here refer to as the local Last Glacial Maximum (lLGM). The ice stream's flow was topographically constrained by the Outer Hebrides, and there is no geological evidence onshore or offshore that the ice stream migrated in position during the glacial cycle (Bradwell et al., 2007). Empirical (Bradwell et al., 2008; Clark et al., 2012) and numerical-model based (Boulton and Hagdorn, 2006; Hubbard et al., 2009) studies show that this wide (~50 km) ice stream drained a large proportion of ice that accumulated over the Scottish Highlands. The MnIS trough can be divided into the outer trough, which is predominantly smooth, with low-strength sediments, and the inner trough with an undulating bed and reduced Early Quaternary sediment cover (Figure 1). The inner trough contains a Neoproterozoic bedrock high, here referred to as the Mid-Trough Bedrock High (MTBH) (Figure 1). Either side of the MTBH, the Minch branches into east and west troughs (Figure 1).

There has been extensive onshore (e.g. Ballantyne and Stone, 2009; Bradwell, 2013) and offshore (e.g. Bradwell and Stoker, 2015; Bradwell et al., 2008; Stoker and Bradwell, 2005) studies of the MnIS catchment. Improved bathymetry data allow the identification of a reverse slope southward of the MTBH. This topography could facilitate MISI, potentially causing rapid retreat south of the MTBH, and would explain the relative sparsity of the landform record between the MTBH and the terrestrial transition (Bradwell et al., 2008; Clark et al., 2018). Numerical modelling also allows for ice sheet processes to be tested that may be less evident in the empirical record, such as the influence of ice shelves. The detailed reconstructions of the MnIS provide an excellent opportunity to test whether ice-sheet models reproduce behaviour recorded by the empirical data.

## 3 Methods

### 3.1 Model description and set-up

We perform numerical simulations with an idealised climate forcing to isolate the effect of marine ice sheet dynamics on the MnIS in order to determine whether the bathymetric set-up of the MnIS could have facilitated MISI. We use the BISICLES ice sheet model (Cornford et al., 2013) to simulate the MnIS. BISICLES is a higher-order ice sheet model with L1L2 physics retained from the full Stokes flow equations (Schoof and Hindmarsh, 2010), which was developed to efficiently and accurately model the dynamics of marine-grounded ice sheets. BISICLES uses adaptive mesh refinement to automatically increase resolution at high velocities and the grounding line, allowing ice stream dynamics and grounding line migration to be well represented, while having a lower resolution in the rest of the ice sheet ensures efficient model speed. Cornford et al. (2013) provides a full description of BISICLES. Although our focus for analysis is the MnIS, we set up the model domain to cover the majority of the British-Irish Ice Sheet (BIIS) at 4 km x 4 km resolution, excluding the central North Sea (Figure 2). Simulating a large portion of the BIIS prevents artefacts caused by domain boundary effects, and allows for migration of ice catchments during deglaciation. For the set-up of the study, we set a 4 km x 4 km grid refined 3 times around the grounding

line in the Minch Sector to produce a maximum horizontal resolution of 500 m x 500 m. The simulations have 10 vertical levels. The friction law uses a linear ($m = 1$) Weertman exponent in accordance with previous BISICLES experiments (Favier et al., 2014; Gong et al., 2017) that were used as the basis for bed friction coefficient map values. We use a calving model that simulates frontal ablation by advecting the calving front with a relative velocity equal to the modelled ice velocity at the front

minus an ablation rate acting in a direction normal to the front. We prescribe a frontal ablation rate of 250 m/y for all lLGM simulations, and 350 m/y for all simulations which prompt deglaciation. These frontal ablation values are prescribed as it allows for ice shelf formation during retreat, a stable extent at the continental shelf edge, and causes only limited deglaciation if Surface Mass Balance (SMB) changes are not included (Figure 4a).

### 3.2 Initial conditions and spin-up

We set the initial conditions to the ice sheet state when the MnIS was as its maximum extent. To avoid the computational costs and uncertainties associated with simulating the full build-up of the ice sheet, we initialise ice thickness in the domain from a perfectly plastic ice sheet model (Gowan et al., 2016) fixed to the lLGM extent at the continental shelf break. The 27ka BP margin of Clark et al. (2012) was used for the remainder of the BIIS. The 27 ka BP margins of the perfectly plastic ice sheet model output match well with the reconstructed maximum extent of the MnIS (Bradwell et al., 2008).

To calculate SMB we use monthly mean surface air temperatures and monthly mean total precipitation from climate model simulations to drive a Positive Degree Day (PDD) mass balance model as in Gregoire et al. (2016, 2015). We use the PDD model PyPDD (Seguinot, 2013), which accounts for the subannual evolution of snow cover, and meltwater refreezing. The PDD factors, and refreezing ratio is summarised in Table 1. The PDD model is driven by mean temperature and precipitation data calculated from the final 50-years of the 26 ka BP bias-corrected equilibrium climate simulation described

by Morris et al. (2018), run with the HadCM3 coupled atmosphere-ocean-vegetation general circulation model (Gordon et al., 2000; Pope et al., 2000; Valdes et al., 2017). This simulation is part of a series of "snapshot" equilibrium simulations covering the last deglaciation that are a refinement of those previously reported by Singarayer et al. (2011) with updated boundary conditions including ice mask, ice orography, bathymetry, and land-sea mask (Ivanovic et al., 2016). It belongs to the same set of simulations used by Swindles et al. (2017) and Morris et al. (2018) to recently derive the climate of the Holocene and

since the Last Glacial Maximum, respectively. We downscale surface air temperatures onto the pre-spin up initial ice sheet surface using a lapse rate of 5.1 K/km, which has been identified as a suitable lapse rate for modelling the Eurasian Ice Sheet (Siegert and Dowdeswell, 2004).

    To remove the effect of a SMB feedback, where surface elevation change causes a positive feedback to SMB due to atmospheric lapse rate, the SMB is decoupled from elevation feedback. In practice, this means that once surface air

temperatures are downscaled onto the initial ice sheet surface elevation to create the SMB map for the domain, SMB does not evolve as the ice sheet surface elevation evolves; there is no lapse rate feedback. This removes the possibility for a SMB instability of retreat, allowing for any MISI during the ice sheet retreat to be isolated.

    We prescribe a sub-ice shelf melt rate using a linear relationship with Sea Surface Temperature (SST);

$$m = -10T$$

Where m is melt (metres of melt water equivalent) and T is the temperature (K) above the freezing point of the ocean, assumed to be -1.8 ℃. This relationship between SST and sub-ice shelf melt rate is based on measurements from Pine Island Glacier, Antarctica (Rignot and Jacobs, 2002). SST values are taken from the same climate simulation that is used to calculate SMB. However, initial SST temperatures are corrected by -2 ℃ as exploratory sensitivity tests show that the uncorrected SST does not allow for ice shelves to form at the lLGM. We correct the SST to permit for ice shelf formation, allowing the influence of the presence or removal of an ice shelf to be tested.

We assume that contemporary land properties were like the bed properties beneath the MnIS. The basal friction coefficient map was produced by grouping regions of similar bed friction, then prescribing values to those regions based on bed friction coefficient values from other studies. Regions of similar basal friction were classified into the following five groups based on observable surface morphological features in satellite imagery and DEMs, and from the glacial map of Britain (Clark et al., 2018) as well as reference to superficial geology maps:

1. Palaeo-ice streams, based upon the presence of mega-scale lineations, convergent flow-patterns from subglacial bedforms and previous reporting in the literature (Margold et al., 2015; Stokes and Clark, 1999). As the main outlets for ice-flow and fastest flow regions, these regions were assigned the lowest basal friction coefficients (β); 100.

2. Marine-sediments, defined based upon geological maps and the presence of characteristic marine bedforms. These are highly deformable, and were therefore assigned a value of 1,000, the second lowest basal friction coefficient (β).

3. Subglacial lineations or drumlins, identified on the glacial map and elevation models. Lineations are thought to represent reasonably fast ice flow and be the product of subglacial bed deformation (Ely et al., 2016). These were assigned an intermediate basal friction coefficient (β) of 2,000.

4. Subglacial ribs or ribbed moraines, identified from previous mapping and elevation models. These are thought to be characteristic of slower ice flow than that of subglacial lineations, and were thus assigned a higher basal friction coefficient. A bed friction coefficient (β) of 3,000 was prescribed here.

5. Exposed bedrock was assigned the highest basal friction coefficient. These high roughness areas were defined by their characteristic surface morphology and from geological maps, and were prescribed the highest value bed friction coefficient (β), 4,000.

The values for bed friction coefficients used as input into BISICLES (Figure 2c) were based on studies of present day ice streams using the same friction law Weertman exponent ($m = 1$), which calculated coefficients by inverting observed surface velocities of Pine Island Glacier and Austfonna Basin 3 (Favier et al., 2014; Gong et al., 2017). This approach simulates an ice stream of a similar morphology as reconstructed using empirical data (Bradwell et al., 2007). Sensitivity tests reveal that the ice sheet volume is sensitive to changes in bed friction coefficient map (Figure 3a), but the extent of ice after a 6,000 year spin-up remains comparable even with different magnitudes of basal friction as ice extent in our experiments is primarily controlled by the continental shelf edge and surface mass balance (Figure 2b).

To recreate isostatically adjusted bed topography, we adjust modern topography using results from a Glacio-Isostatic Adjustment (GIA) model (Figure 2d). GEBCO (Becker et al., 2009) provides modern offshore bathymetry, and SRTM (Farr et al., 2007) provides onshore topography. Isostatic adjustment uses results from the EUST3 GIA model (Bradley et al., 2011). EUST3 accounts for near-field and far-field isostatic adjustment due to ice loading. The Relative Sea Level (RSL) change from EUST3 at 30ka BP is used to deform contemporary topography, maintaining a high resolution ice sheet bed whilst also accounting for RSL change.

### 3.3 Experimental design

We designed our experiments to use an idealised representation of the external forcings of ice sheet retreat (surface mass balance and sub-shelf melt) in order to isolate the internal ice sheet mechanisms and instabilities of retreat. All experiments begin at a stable lLGM volume, with continental shelf edge glaciation and a small ice shelf (<4 km from shelf front to the grounding line) (Figure 3). Experiment set-up is summarised in Table 2. These experiments test the applicability of MISI to the MnIS. In reality, ice sheet evolution is a function of climate fluctuations, ice surface evolution, and sea level change as well as MISI. However, to isolate the effects of MISI, sea level is held constant throughout the experiments, there is no elevation-SMB feedback, and the climate change is a simple step-change.

### 3.3.1 Deglaciation

The end of the *SPINUP* simulation is used as the start point for the *RETREAT* experiment. In the simulation *RETREAT,* we use an idealised climate perturbation approach to trigger deglaciation, which consists of applying an instantaneous uniform warming of the surface air temperature by 1.5 K and SST by 2 K each month, without changing precipitation. The magnitude of these perturbations are based on changes between the 26 and 18 ka BP equilibrium climate simulations. The resulting surface mass balance and sub-ice shelf melt are then kept constant throughout the run. As for the *SPINUP* simulation, SMB is decoupled from an ice elevation-SMB feedback. Therefore, any change in the rate of ice sheet retreat is caused by internal ice sheet dynamics.

To test the relative role of ocean and atmospheric warming in driving the retreat, we ran *RETREAT_ATMOS* and *RETREAT_OCN* where only SMB or sub-ice shelf melt rate are perturbed respectively. Both of these simulations only lead to partial deglaciation (Figure 4a), while the combination of both forcings (*RETREAT*) cause deglaciation to the northwest Scottish Highlands. We keep bed topography and sea level constant for the duration of the deglaciation simulations as an evolving sea level could interact with the process of MISI. Therefore, a constant sea level is necessary to understand the potential for MISI during the retreat of the MnIS. In reality, only a small RSL change (~15 m sea level fall) would be expected because of the competing effects of BIIS expansion and global sea levels fall (Bradley et al., 2011).

### 3.3.2 Reversibility of ice stream retreat

We test whether retreat is irreversible once it is initiated, or whether ice volume and area recovers to lLGM levels given a return to lLGM climate. In experiment *READVANCE,* we test for the reversibility of ice stream retreat by reverting the climate perturbations during the deglaciation. A set of simulations were started from points at 800 years intervals through the *RETREAT* simulation with the boundary conditions returned to lLGM and run for 10,000 model years to allow the ice sheet to reach a new stable state (constant volume and extent). Here, ice sheet collapse is defined as an ice sheet not returning to its lLGM extent given lLGM boundary conditions following retreat.

### 3.3.3 Ice shelf influence

We tested whether an ice shelf is important in influencing the dynamics of the MnIS retreat, by providing a buttressing force to reduce ice stream flux over the grounding line. To test this hypothesis, we ran a simulation (*RETREAT_NOSHELF*) where the sub-ice shelf melt rate was increased to 100 m/y (52.9 m/y higher than during the *RETREAT* simulation) in order to force the removal of the ice shelf during deglaciation. Note that, in this experiment, we kept the frontal ablation rates identical to *RETREAT* in order to isolate the effects of ice shelf buttressing, removing the influence of any increased ocean ice mass loss.

### 4 Results and Discussion

### 4.1 Pattern of retreat

Imposing a constant SMB and sub-ice shelf melt perturbation (simulation *RETREAT*) causes a retreat of the MnIS from the continental shelf edge to the highlands within 8,000 model years (Figure 4; Video S4). Although the SMB and sub-ice shelf melt are constant through the simulation, with SMB decoupled from the change in surface elevation, the rate of volume and area change fluctuates through the simulation (Figure 4a), whilst an exponential decay in ice volume would be expected given a simple climate forcing (Harrison et al., 2003; Johannesson et al., 1989; Nye, 1963, 1960). The evolution of the ice stream during the *RETREAT* simulation can be divided into three stages; an initial retreat phase, a stagnation phase, and a late retreat phase. Volume loss is most rapid at the initial retreat phase (Figure 4b-d), reducing the domain's ice volume in the first 2,000 model years by ~25%. In the stagnation phase, between 2,000 and 6,300 model years of the simulation (Figure 4d-e), volume loss slows. Finally, during the late retreat phase volume loss slightly increases from 6,300 model years onwards (Figure 4e-f). The slower rate of volume loss in the stagnation phase occurs as the margin retreats beyond marine influence onto the Outer Hebrides for the majority of the domain. At this stage the ice stream is in a similar marine setting to many tidewater glaciers in Greenland and Antarctica (Joughin et al., 2008), and the re-acceleration of the late retreat phase only begins once the grounding line has retreated further south towards the inner trough of the Minch (Figure 4e).

The ice area loss produces a broadly similar trend to volume loss, although in the stagnation phase (3,500 to 6,300 model years) there is negligible area loss despite continuing volume loss (Figure 4a). This is associated with a near-stagnation of the grounding line when the margin of the ice sheet has mostly retreated onshore and the marine-terminating ice stream becomes confined between the Scottish mainland and the Outer Hebrides (Figure 4d-e). During the stagnation phase ice loss occurs through thinning with limited change in margin position. As the thinning continues, the ice area begins to retreat more rapidly just after 6,300 model years, which coincides with the start of a rapid margin retreat in the Minch (figure 4e).

## 4.2 Role of ice shelf buttressing

Ice shelves provide a buttressing membrane stress to the ice streams flowing upstream of them (Hindmarsh, 2006). Ice stream acceleration in response to the sudden collapse of ice shelves has been observed in the contemporary Antarctic Peninsula (Scambos et al., 2004). The buttressing effect of ice shelves can also allow a stable grounding line position given a reverse sloping bed (Gudmundsson, 2013), meaning it is important to consider the impact of ice shelves when examining MISI. The ice shelf that forms during the simulations of the MnIS is initially unconstrained by topography or surrounding ice until the grounding line retreats during the stagnation phase of retreat at 6,300 years (Figure 4e). It is therefore expected that removing the ice shelf from the simulations whilst triggering deglaciation will initially have negligible impact on the retreat of the ice stream, before having greater impact once the ice shelf is constrained by surrounding grounded ice.

For the first 5,000 model years the simulated grounded ice volumes diverge by less than 1% in the simulations with (*RETREAT*) and without (*RETREAT_NOSHELF*) an ice shelf (Figure 5). After 5,000 model years there is a notable divergence in the evolution of grounded ice volume of the two simulations, where the simulation without an ice shelf has higher rates of mass loss compared to the simulation with ice shelves (Figure 5a). At the time of maximum volume difference between the simulations (7,000 model years), the simulation *RETREAT_NOSHELF* has a grounded volume ~2,000 km$^3$ (10%) smaller than the simulation *RETREAT*. However, the acceleration of mass loss during the late retreat phase occurs in both simulations, suggesting the presence or removal of an ice shelf cannot prevent continued ice volume loss. Although the presence of an ice shelf affects the ice volume, it has almost no impact on the location of the grounding line and its chronology of retreat (e.g. Figure 5b-c), with the difference in the grounding line location mostly within the finest resolution of the model (500 m). The removal of the ice shelf also means that the grounding line has no mechanism for stability on the reverse sloping bed retreat southward of the MTBH (Gudmundsson, 2013). These results suggest that the presence or absence of an ice shelf has only a limited influence on the spatial pattern and timing of MnIS retreat. A significant difference in ice volume is not reflected in grounding line position, showing that ice flux across the grounding line is important, not just the position of the grounding line. The significance of the ice shelf influence demonstrates that in order to be confident of the future evolution of our contemporary ice sheets, we need to be confident of the future evolution of ice shelves over centennial timescales.

The unconstrained nature of the Minch Ice Shelf at the start of the deglaciation makes the mechanics of ice shelf buttressing different from many areas of the contemporary West Antarctic Ice Sheet, where large ice shelves are buttressed laterally by surrounding ice or bedrock (Pritchard et al., 2012). In a number of prominent places in Antarctica, ice shelves are

also pinned from underneath by bedrock rises (Matsuoka et al., 2015). This may suggest that the dynamics of the Minch Ice Shelf during the first stage of retreat are more analogous to East Antarctic Ice Shelves. The Minch Ice Shelf is not constrained laterally until the later stages of the deglaciation where the shelf is supported by surrounding ice (Figure 5), where the topographic setting is similar to examples of fjord-like confined glaciers in Greenland (Joughin et al., 2008). Whilst the MnIS and many Greenland ice streams can retreat beyond direct marine influence (Funder et al., 2011), this is not possible for Antarctic ice streams grounded in deeper troughs. Empirical evidence of the last Eurasian ice sheet has the western ice margin constrained by the continental shelf edge across the majority of the Atlantic margin (Hughes et al., 2016). This suggests that ice shelves at maximum ice extent across the Atlantic margin would be similar to the Minch Ice Shelf in the early stages of deglaciation, limited in size by the continental shelf break, and unconstrained. It is likely, therefore, that ice shelves along this Atlantic margin were not influential in the retreat of the ice sheet until the grounding line retreated to areas laterally constrained by topography or surrounding ice.

### 4.3 Testing ice stream instability with readvance experiments

Given the bathymetric profile of the ice stream path (Figure 6), it would be expected that the MnIS would experience MISI ~180 km along the AB transect (Figure 6b). This is because of the reverse slope (downhill retreat) that the ice stream path would encounter in the later stages of the deglaciation after retreating beyond the MTBH. The experiment *READVANCE* returns the ice stream to the lLGM climate at 800 year intervals during the *RETREAT* experiment, in an attempt to recover the ice stream to lLGM extent.

For the first 5,600 model years of the recovery experiment (*READVANCE*) ice volume returns to lLGM extent given a return to lLGM forcings. All simulations started from *RETREAT* at model year 800 to 5,600 readvance to the lLGM extent. However, in simulations with a point of recovery (initial conditions) beyond 5,600 model years, the ice stream does not recover to lLGM extent given lLGM boundary conditions (Figure 7); instead it evolves towards a reduced stable state with a volume ~25% smaller, and an area ~50% smaller than the stable lLGM state (Figure 7a,b).

We identify two stable ice conditions in these *READVANCE* simulations given our initial lLGM forcings; a full shelf-edge glaciation (Figure 7c), and a small Hebrides Ice Cap with glaciation in the Minch limited to the east trough (Figure 7d). The resulting stable extent of the ice stream is dependent on the evolution history of the ice stream; there is hysteresis in the MnIS evolution. Hysteresis of ice sheet evolution suggests an instability during the advance or retreat of an ice sheet (Schoof, 2007).

The zone of collapse is defined here as the point of divergence of recovery states; the position of the grounding line at which returning to lLGM forcing no longer allows recovery to initial lLGM extent, and only allows recovery to the new stable condition of limited Minch glaciation (Figure 7d). The zone of collapse occurs 5,600-6,400 model years into the retreat, ~180 km along the mapped AB transect, as the margin retreats onto a retrograde bed slope (Figure 6). The point of collapse is simulated to occur after the margin has retreated back from the MTBH, suggesting that the MnIS transitions through a zone of instability at this point, thus indicating that the observed hysteresis is caused by MISI.

As well as the influence of MISI, the morphology of the ice stream marine margin may also cause an instability during the retreat of the ice stream. For the majority of the retreat, the ice stream is buttressed to the east and west by surrounding ice (Figure 4). However, in the smaller stable state (Figure 7d), the ice stream is not buttressed on the western margin, due to the bay forming in the west trough of the Minch. This removal of the lateral buttressing of the ice stream to the west may also contribute to the inability of the ice stream to recover the lLGM extent in this experiment.

The difference between the grounding line position at the zone of transition and at the collapse state, i.e. the magnitude of collapse, varied across the ice stream (Figure 8a). In the East Trough the magnitude of collapse was limited, and a bathymetric transect shows only a limited reverse slope to facilitate MISI (Figure 8b). However, in the West Trough the magnitude of collapse was more significant, and the bathymetric transects shows a more sustained reverse slope to facilitate MISI (Figure 8c).

With ice unable to advance beyond the MTBH in the recovery simulations, a mechanism is required to allow ice to first advance towards the shelf edge prior to the lLGM, but then be unable to advance after retreat from lLGM extent. Although this study started with an ice extent already at the lLGM extent, other studies such as Patton et al. (2016) successfully simulated the build-up of the entire Eurasian ice sheet, with ice overcoming the MTBH and reaching the shelf edge given an lLGM climate. The variability and uncertainty in the climate forcing could allow readvance back to the continental shelf edge. In particular, in our model, if the lLGM climate is uniformly cooled by 0.5 K across the entire domain ice readvances from a "collapsed" position to an lLGM position. It is reasonable that this small correction falls within the error of the climate simulations or variability in the climate during the build-up phase. Nonetheless, even with the cooled climate, recovery back to lLGM extent is considerably slower beyond 5,600 model years. Despite cooling the climate to force a recovery to lLGM extent, a behavioural change in the ice stream readvance beyond 5,600 model years remains.

Topographic changes over the course of the glacial cycle may also explain why the simulations did not readvance over the MTBH given lLGM conditions. These topographic changes would be present in two ways; changes in isostasy during the glacial cycle, and bed erosion and deposition. The simulations run on an ice sheet bed adjusted for isostasy at 30ka BP, which exaggerates the reverse slope causing MISI. The original advance of the ice stream would likely be on a bed where the slope was reduced. Bed erosion could also increase the prominence of the hard bedrock topographic high after initial glaciation. According to the modelling of Patton et al. (2016), the MnIS is an area with high potential cumulative erosion 37-19ka BP. Both these mechanisms could exaggerate the reverse slope subsequent to glaciation, potentially allowing initial glacial advance, but not readvance from a retreated state. Therefore, this demonstrates the importance of understanding future topography evolution in understanding the long-term evolution of out contemporary ice sheets.

**4.4 Comparison to empirical reconstructions**

To investigate the mechanism of marine influence on the MnIS, and simplify the experimental design, several assumptions were made which take the experiments away from "reality". In reality, the various controlling factors of ice sheet change would interact together to exaggerate or dampen the effect of MISI. For example, rebound of the ice sheet bed during

deglaciation could dampen or eliminate the observed effect of MISI. Similarly, an elevation-SMB feedback could exaggerate any observed MISI as the ice sheet thins and retreats. Removing the experiments from a set-up akin to reality is justified to investigate the possibility of marine influence on the BIIS. The simulations test the applicability of MISI to such a bathymetric setting, but do not imply a relative importance of MISI. To consider relative importance, the simulations would also need to consider evolving sea level, elevation-SMB feedback, evolving climate, thermodynamics, and bed friction, amongst others. This approach remains beyond current ice sheet models.

The resulting pattern of retreat from the *RETREAT* simulation has a number of similarities with previous retreat reconstructions (Bradwell et al., 2008; Clark et al., 2012; Hughes et al., 2016) despite the idealised nature of the climate forcing. For example, the margin appears to "hinge" on the northern point of the Isle of Lewis (Bradwell et al., 2007; Bradwell and Stoker, 2015). The margin also recesses into the Minch, with a small ice cap on the Outer Hebrides persisting during deglaciation. The idealised nature of the climate forcing means that pattern and relative rate of retreat can be compared to empirical reconstructions, but the absolute timing cannot. There are close similarities between the simulated retreat and reconstructed retreat in the later stages of the deglaciation. In the later stages of retreat, both in the reconstructed and simulated retreat, the east trough contains a small ice stream whilst the west trough has fully deglaciated and formed a calving bay (Bradwell and Stoker, 2015). We take the key similarities of retreat pattern and margin morphology between the simulated deglaciation and empirical deglaciation reconstructions as an indication that the main mechanisms that controlled the deglaciation of MnIS are replicated in our simulations.

All simulations in this study use constant climate forcing which do not account for the evolution of climate during the deglaciation of the BIIS. The fluctuations in volume and area change in the simulations are the signal of internal dynamical ice sheet processes, which do not include the feedback between SMB and elevation or any transient evolution of external forcing in climate and sea level.

The empirically reconstructed retreat history of the MnIS is the result of both the ice sheet internal and external forcing signal. Therefore, the signals simulated in these experiments could be exaggerated or dampened by external forcing; the climate signal. This style of retreat is evident from the moraine record (Figure 1), which shows a series of back stepping moraines across the continental shelf (Bradwell and Stoker, 2015; Bradwell et al., 2008; Clark et al., 2018). It is inferred that these moraines form during a period of relative stability of the margin. These large seabed moraines occur in several places across the shelf edge, but there is only one period during the *RETREAT* simulation when the margin is stable, when the margin passes the northernmost tip of Lewis and enters the Minch (Figure 4 a and d). The surface expressions of the large seabed moraines are present in the bed topography used in the simulations. However, the ice margin retreats over other areas with a greater surface expression than these morainic wedges with no stabilisation of the margin.

The area and volume loss acceleration during the late retreat phase, which we interpret as the beginning of MISI, in reality would also be influenced by the climate fluctuations. The magnitude of this area and volume loss would be exaggerated or dampened depending on the increasing or decreasing SMB of the ice sheet at any given period. Overall, the simulated volume and area change in these simulations is relatively smooth, whilst in reality climate fluctuations will have caused annual

or even decadal dynamic margin fluctuations, more akin to the Hubbard et al. (2009) simulation of the ice sheet evolution which was driven by a transient climate.

Geomorphological studies of palaeo ice sheets are able to reconstruct the area of ice sheets. Reconstructing ice thickness, and therefore volume, requires the introduction of further uncertainties, like bed friction and SMB, and therefore it is sometimes assumed that area change can directly inform volume change of the ice sheet (Hughes et al., 2016). However, model simulations of palaeo ice sheets can consider both area and volume changes. In these simulations although volume and area evolution showed a similar trend, the area evolution during *RETREAT* had more obvious pauses and accelerations in area loss than the pattern of volume loss. A change in ice stream velocity could also alter ice sheet volume without altering the ice sheet area, as shown by varying the bed friction in the *SPINUP* experiment (Figure 3a). In this example, understanding the area change through geomorphic empirical evidence may overstate the magnitude of changes expected in volume loss.

An idealised climate warming perturbation was used to trigger deglaciation in the simulations (e.g. *RETREAT*) in order to isolate the internal mechanisms of retreat. However, the retreat began before ~27ka BP, when the NGRIP δ18O record suggests a cooling rather than a warming climate over Greenland (Andersen et al., 2004). Whilst rapid deglaciation can be achieved in ice sheets with significant ice streaming due to ice stream acceleration (Robel and Tziperman, 2016), a climate trigger is required to increase the surface slopes at the ice sheet margin. Model results also suggest a generally cooling global climate at ~27ka BP (Singarayer and Valdes, 2010). The warming mechanism we used to trigger deglaciation in our simulations is not apparent in the climate record. The results of the *RETREAT* simulations also indicate that internal dynamical mechanisms alone would not have been sufficient to continue retreat until the triggering of MISI when the ice stream retreats past the MTBH. It seems that the MnIS was retreating in a cooling climate, but these simulations idealised climate forcing to isolate internal ice sheet instabilities, and therefore do not reveal a mechanism that explains MnIS retreat in a cooling climate. There are two likely possibilities that could explain the MnIS retreating in a cooling climate (explained below); i) internal mechanisms of the BIIS, or ii) a local SMB change of this sector of the ice sheet prior to ~27ka BP.

A local SMB change could be caused by atmospheric warming or a reduction in precipitation. Local warming of the northern British Isles, whilst the rest of the BIIS and other northern hemisphere ice sheets expanded, could be a mechanism to explain early MnIS retreat. The NGRIP ice core is more than the synoptic scale away from the Minch, and there could reasonably be local warming during global cooling. Given the uncertainty in palaeo climate modelling, particularly at high latitude due to sensitivity to ice-sheet forcing (Singarayer and Valdes, 2010), it is a reasonable possibility that a local warming would not be included in the HadCM3 last glacial simulations used to force the ice sheet model. However, it seems unlikely that climate could be warming in the Minch whilst allowing rapid expansion for the remainder of the BIIS. Additionally, a local reduction in precipitation could be caused by a change in the position of the polar front, which would have been affected by the advance and retreat of the northern hemisphere ice sheets during the last glacial cycle (Oster et al., 2015). A migration in the polar front would likely affect the entire western margin of the BIIS, potentially causing the start of deglaciation to be asynchronous with the rest of the northern hemisphere (Scourse et al., 2009). However, as the forcing in these experiments remains idealised, this is only a speculative driver of early BIIS retreat.

Alternatively, internal mechanisms of the BIIS that were not simulated as part of this study could explain a retreating MnIS in a cooling climate. Two candidates for this mechanism are ice piracy from other catchments of the BIIS, or the initial advance of the MnIS being the result of a surge-type advance. The simulations presented here did not feature a lapse rate effect on SMB as the ice sheet surface lowered into a warmer climate. This process could help facilitate rapid retreat in a cooling climate, but could not be a trigger for retreat. During the retreat of the MnIS, ice was extending over Northern Ireland out to the shelf edge (Clark et al., 2012; Dunlop et al., 2010). It is theorised that a significant proportion of the ice feeding this advance came from the Hebrides Ice Stream, evidenced by large moraines northward of Donegal Bay. The source areas of the Hebrides Ice Stream will have significantly overlapped with the MnIS, and the advance of the Hebrides Ice Stream could have initiated ice piracy from the MnIS, causing retreat in a cooling climate. The initial advance of the MnIS could also have been caused by a surge-type advance, which subsequently retreated as the remainder of the ice sheet advanced. These experiments assumed an initial steady state at lLGM extent. Dating the advance of ice sheets is inherently more uncertain than retreat (Hughes et al., 2016), and although the significant Trough Mouth Fan could be taken as evidence for a relatively stable MnIS, it almost certainly formed over multiple glacial advance and retreat cycles (Bradwell and Stoker, 2015). These mechanisms are speculative, and would require experiments with transient forcings to test. These internal instabilities would not have been represented in these simulations because a warming climate was used to trigger deglaciation, forcing the Hebdrides Ice Stream to retreat rather than advance. Bed friction and topography also do not evolve during the simulation, meaning ice streams in the simulations cannot activate or shutdown due to sediment exhaustion, or bed hydrology changes. Bed hydrology evolution has been identified as a key control of ice streams, but this process remains challenging to incorporate into ice sheet models (Hewitt, 2013). These simulations therefore provide evidence for internal processes during the retreat of the MnIS, but the initial trigger for deglaciation of the MnIS in a cooling climate remains elusive.

## 5 Conclusions

We simulated the retreat of the MnIS from a position of maximum extent, using an idealised climate perturbation in order to identify the role of the internal dynamical mechanisms on ice sheet retreat. This simulation showed a retreat in three phases, an initial retreat (0-3,500 model years), stagnation (3,500-6,300 model years), and a late retreat (after 6,300 model years). The stagnation phase occurred as the ice stream retreated past the northernmost tip of the Isle of Lewis. This slowing of volume and area loss coincides with a significant proportion of the ice sheet margin retreating onto the Outer Hebrides, beyond marine influence. At this point in the simulation, the MnIS is in a similar marine setting to many retreating tidewater glaciers in Greenland and Antarctica. During this phase, a laterally constrained ice shelf provided buttressing to the ice stream. In the late retreat phase, the area and volume loss rate re-accelerated as the grounding line retreated on a reverse bed slope. We reversed the idealised climate perturbation regularly during this simulated deglaciation to test for instabilities of retreat. After the re-acceleration of volume and area loss at 6,300 model years, the ice sheet did not recover to lLGM volume and area given lLGM conditions. We suggest this result is evidence for a retreat of the MnIS caused by marine ice sheet instability.

We compared the simulated retreat to a simulation with the ice shelves removed, which was otherwise identical. The simulations show that ice shelves were not influential to the magnitude and pattern of retreat for the first 5,000 model years of the simulated deglaciation, when the ice shelf was unconstrained. Once the margin had retreated into a trough geometry that constrained the ice shelf, the removal of the ice shelf caused higher ice stream velocities and a more rapid ice volume loss. We therefore find evidence for an influential ice shelf buttressing effect during MnIS deglaciation. Our simulations demonstrate the importance of MISI and ice shelf buttressing during retreat of the Minch ice stream. These processes currently represent the largest source of uncertainty in projecting the future evolution of the Antarctic ice sheet. We suggest that the detailed chronology of BIIS retreat currently being produced by the BRITICE-CHRONO project has the potential to constrain important processes controlling MISI in models used for future sea level projections.

*Code and data availability*. We used a branch of the BISICLES ice sheet model, revision 3635 (https://anag-repo.lbl.gov/svn/BISICLES/public/branches/smb/). Output files and required input files to reproduce the described experiments can be found at (https://doi.org/10/cvv7). Code for the PDD model used is available at (https://github.com/juseg/pypdd).

*Competing interests*. The authors declare that they have no conflict of interest.

*Acknowledgements*. NG was funded by a studentship from the Natural Environment Research Council (NERC) SPHERES Doctoral Training Partnership (NE/L002574/1) with CASE support from the British Geological Survey. This work was undertaken on ARC3, part of the High Performance Computing facilities at the University of Leeds, UK. JCE, CDC and TB were funded by the NERC consortium grant; BRITICE-CHRONO NE/J009768/1. RFI was supported by a NERC Independent Research Fellowship (NE/K008536/1). This work has benefited from extensive discussions with the BRITICE-CHRONO consortium team. We thank Julien Seguinot and one anonymous reviewer for their thoughtful and constructive feedback that has improved the clarity of the manuscript.

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

| Parameter | Value | Units |
|---|---|---|
| Flow law | Power Law | - |
| Weertman exponent | 1 | - |
| Ice temperature | 268 | K |
| Ice density | 918 | kg m$^{-3}$ |
| PDD$_{ice}$ factor | 0.008 | m °K$^{-1}$ d$^{-1}$ |
| PDD$_{snow}$ factor | 0.003 | m °K$^{-1}$ d$^{-1}$ |
| Re-freeze ratio | 0.07 | - |
| Snow-rain threshold | 275.15 | K |
| Lapse rate | 5.1 | K/km |

**Table 1: Key model variables and parameters.**

| Name | Initial ice Thickness | Surface mass balance | Sub-shelf melt | Basal Friction |
|---|---|---|---|---|
| *SPINUP* | Plastic thickness | 26 ka BP | 26.3 m/y | Standard (Fig. 4a) |
| *RETREAT* | *SPINUP* | 26 ka BP + 1.5 K | 47.1 m/y | Standard |
| *RETREAT_ATMOS* | *SPINUP* | 26 ka BP + 1.5 K | 26.3 m/y | Standard |
| *RETREAT_OCN* | *SPINUP* | 26 ka BP | 47.1 m/y | Standard |
| *READVANCE_xxxxyr* | Retreat at model *x* yr (800 yr intervals) | 26 ka BP | 26.3 m/y | Standard |
| *RETREAT_NOSHELF* | *SPINUP* | 26 ka BP + 1.5 K | 100 m/y | Standard |
| *READVANCE_COOLING* | Stable READVANCE_8000yr | 26 ka BP − 1.5 K | 26.3 m/y | Standard |
| *SPINUP_MAXFRICT* | Plastic thickness | 26 ka BP | 26.3 m/y | Standard x 1.5 |
| *SPINUP_MINFRICT* | Plastic thickness | 26 ka BP | 26.3 m/y | Standard x 0.5 |

10    **Table 2: Summary of experiment set-up and forcing**

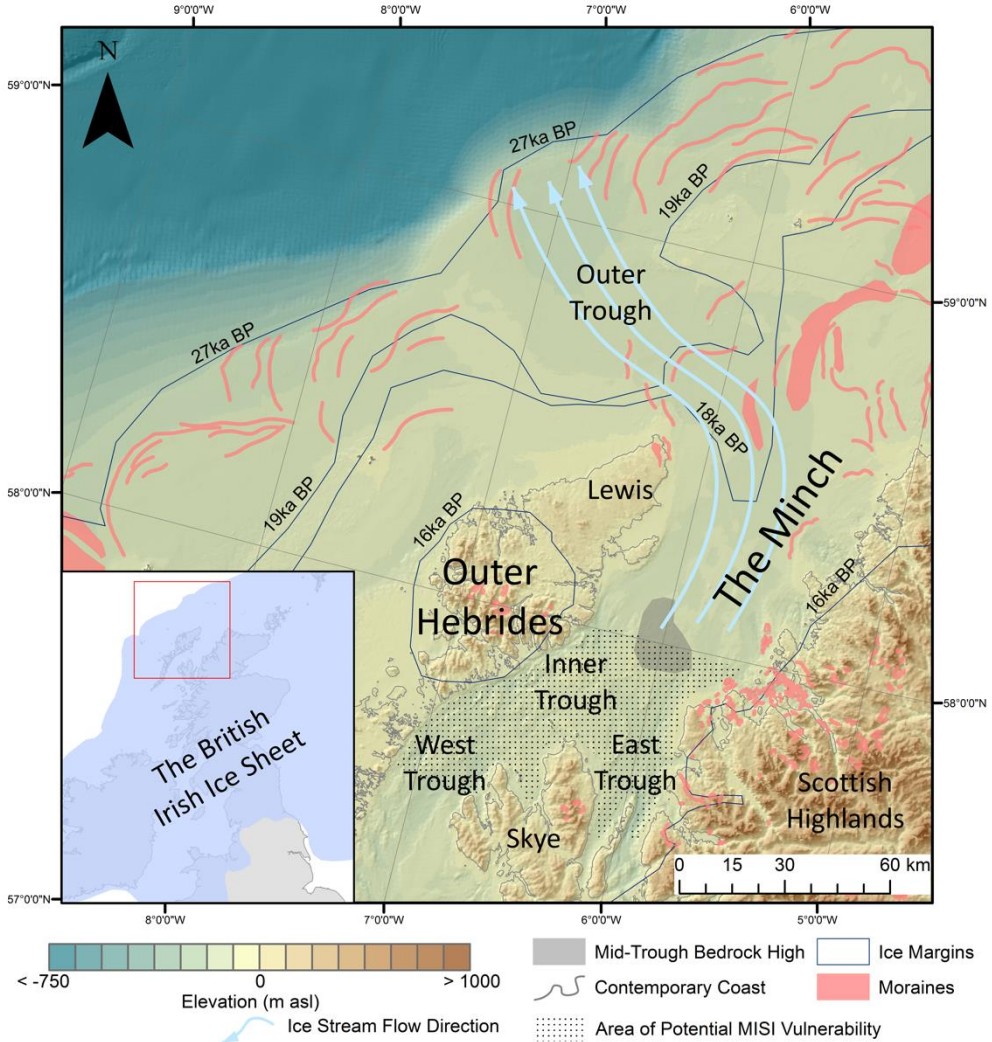

**Figure 1: Location and geographical setting of the Minch Ice Stream. The inset shows the maximum extent of the British-Irish Ice Sheet from Clark et al. (2012). The red box indicates the Minch ice stream region shown in the larger map. Ice margins and dates are from Clark et al. (2012). Moraines are from the BRITICE glacial landform map, version 2 (Clark et al., 2018). Area of potential MISI vulnerability is inferred from the presence of marine retrograde slopes. Key locations mentioned in the text are labelled.**

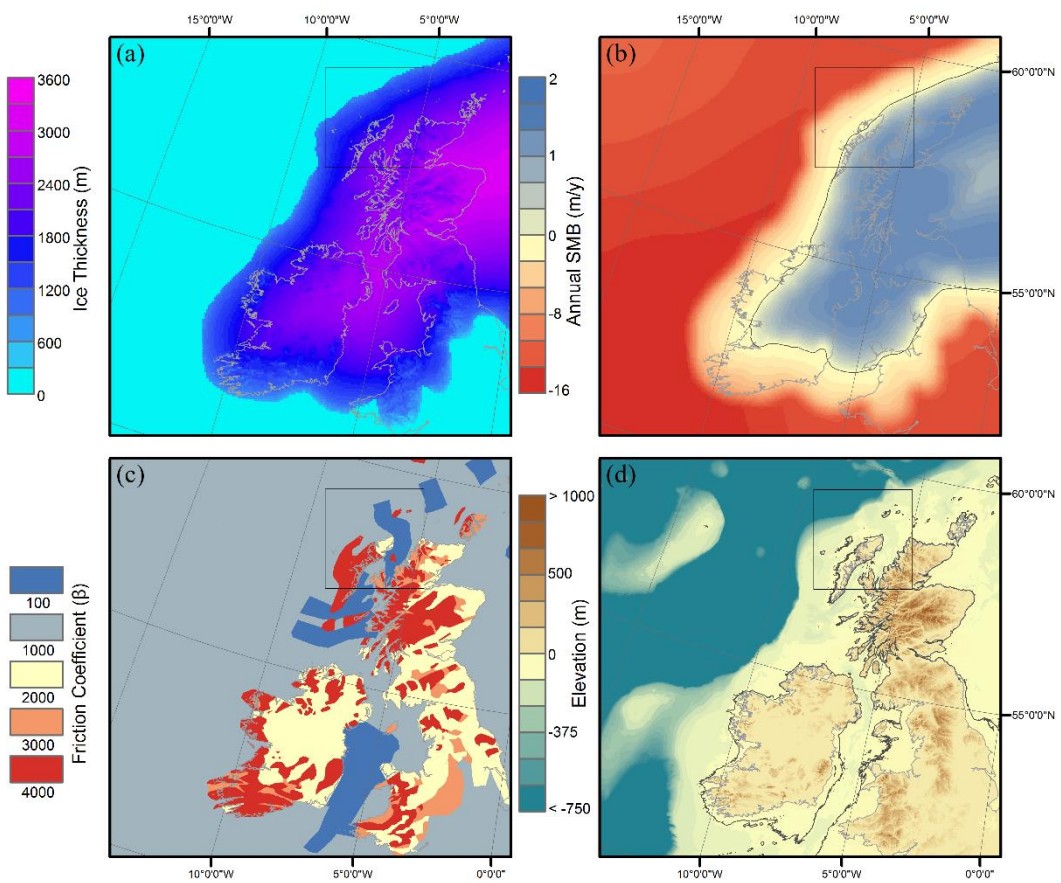

**Figure 2: Initial conditions and boundary conditions for the equilibrium spin-up lLGM ice sheet simulation, showing the full domain of the simulations. (a) Initial ice thickness (m), (b) Annual Surface Mass Balance (m/y), with black contour line representing the Equilibrium Line (SMB = 0 m/y). (c) Bed Friction Coefficient (β), d) Isostatically adjusted topography (m) corresponding to 30 ka BP. The maps show the full ice sheet domain and the black boxes indicate the area of model grid refinement referred to as the Minch sector.**

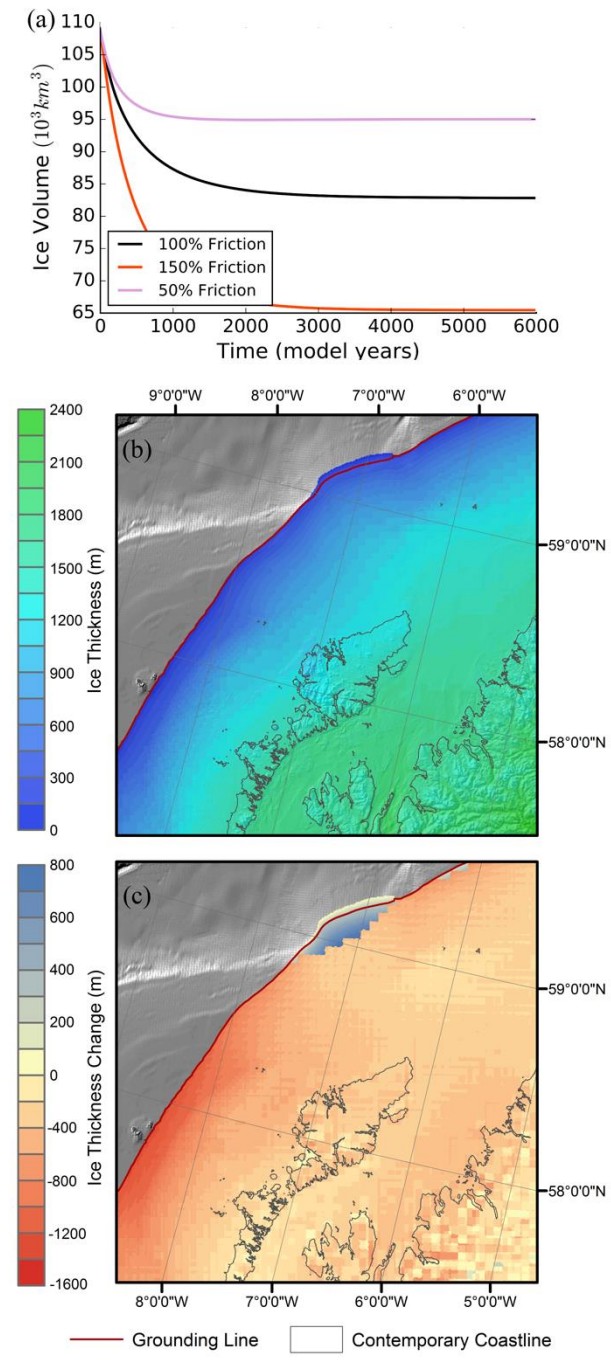

**Figure 3: a) Evolution of the ice sheet volume over the Minch sector during spin-up. b) Simulated ice sheet thickness after 6,000 model years of spin-up, c) Ice thickness change between the start and end (year 6000) of the spin-up simulation (i.e. difference between Figure 2a and Figure 3b.)**

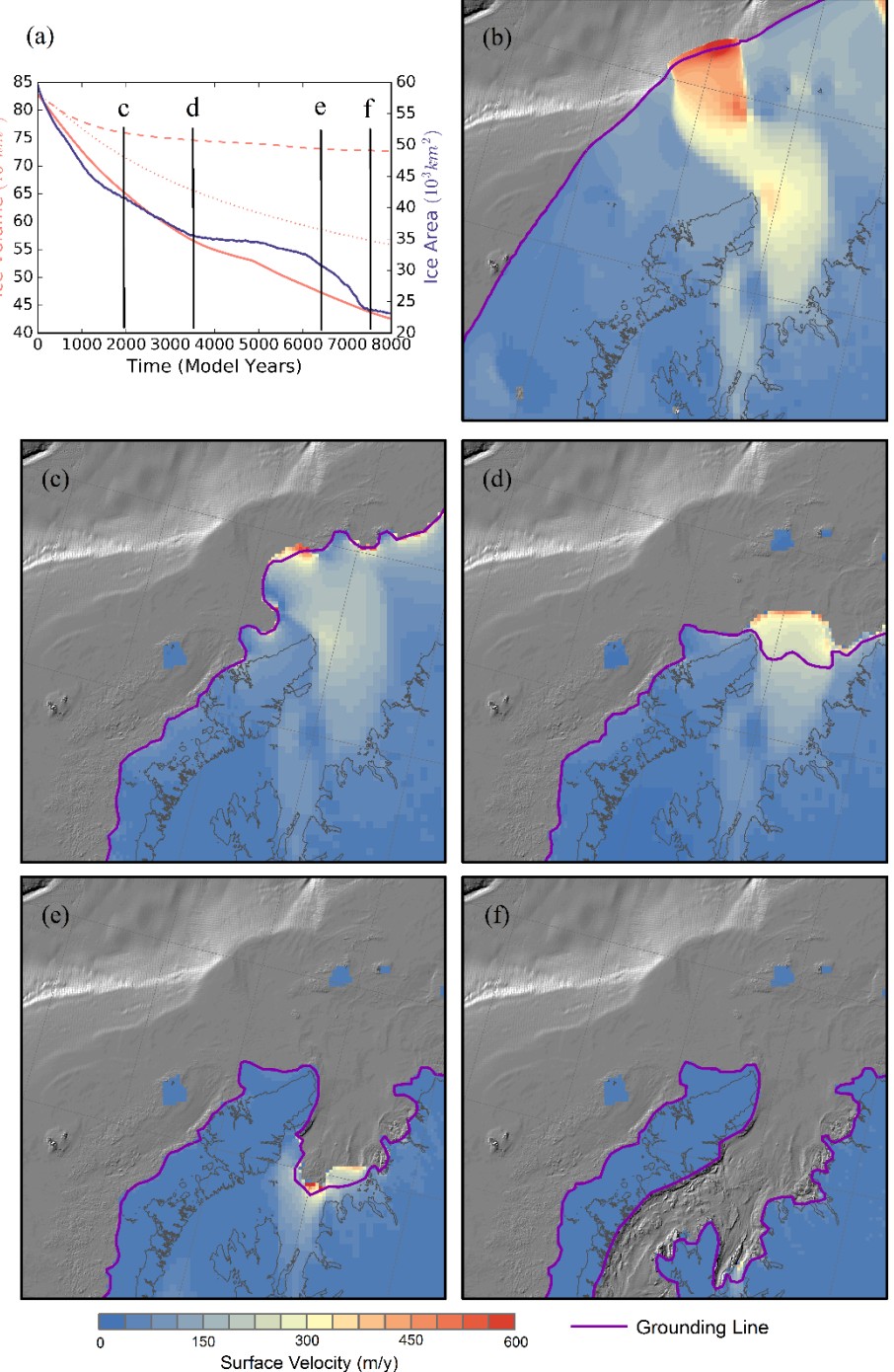

**Figure 4: Evolution of the ice sheet in the Minch sector in the *RETREAT* simulation. a) Timeseries of ice volume and area over the Minch sector. The dashed curve shows the volume response to *RETREAT_OCN*. The dotted curve shows the response to *RETREAT_ATMOS*. Ice surface velocity (b, c, d, e, and f) at 0, 2000, 3500, 6300, and 7500 model years respectively with grounding line shown in purple.**

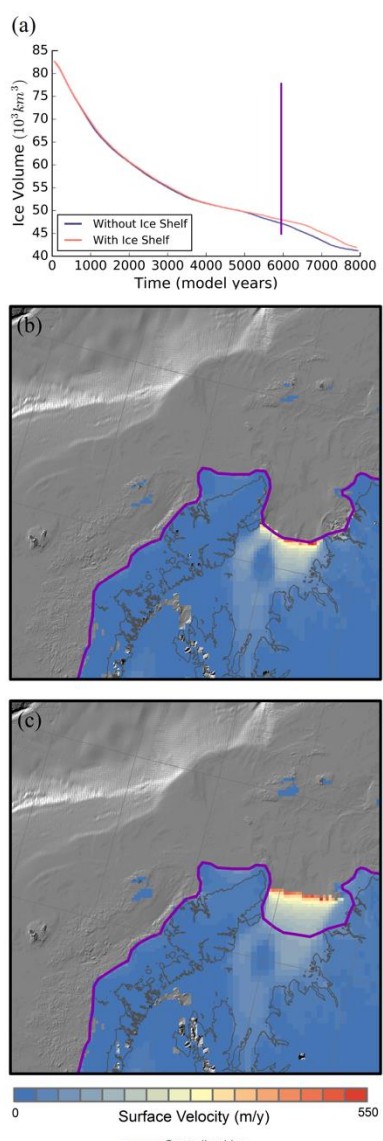

**Figure 5: Effect of ice shelves. (a) Evolution of grounded ice sheet volume over the Minch sector in simulation *RETREAT* with ice shelves (red line) and *RETREAT_NOSHELF* in which ice shelves are forcibly removed (blue line). (b,c) Surface velocity and grounding line location (purple line) in RETREAT_NOSHELF (b) and *RETREAT* (c).**

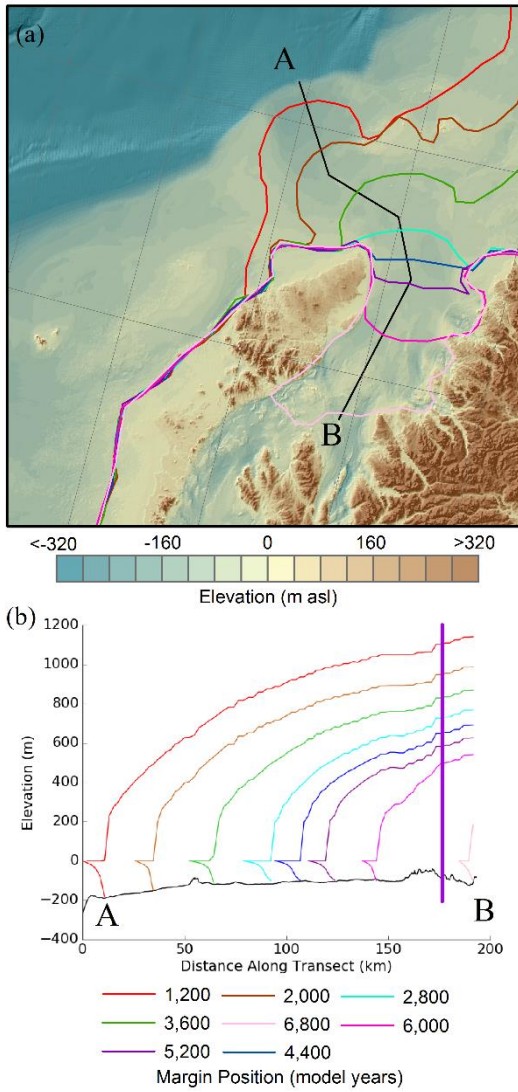

**Figure 6: a) The grounding line position of the ice sheet at 800 year intervals during the *RETREAT* simulation. b) Transect through the ice stream showing the ice sheet elevation at the same intervals. The purple line marks the grounding line position in the stable "collapsed" ice sheet state from the *READVANCE* simulations (Figure 7d).**

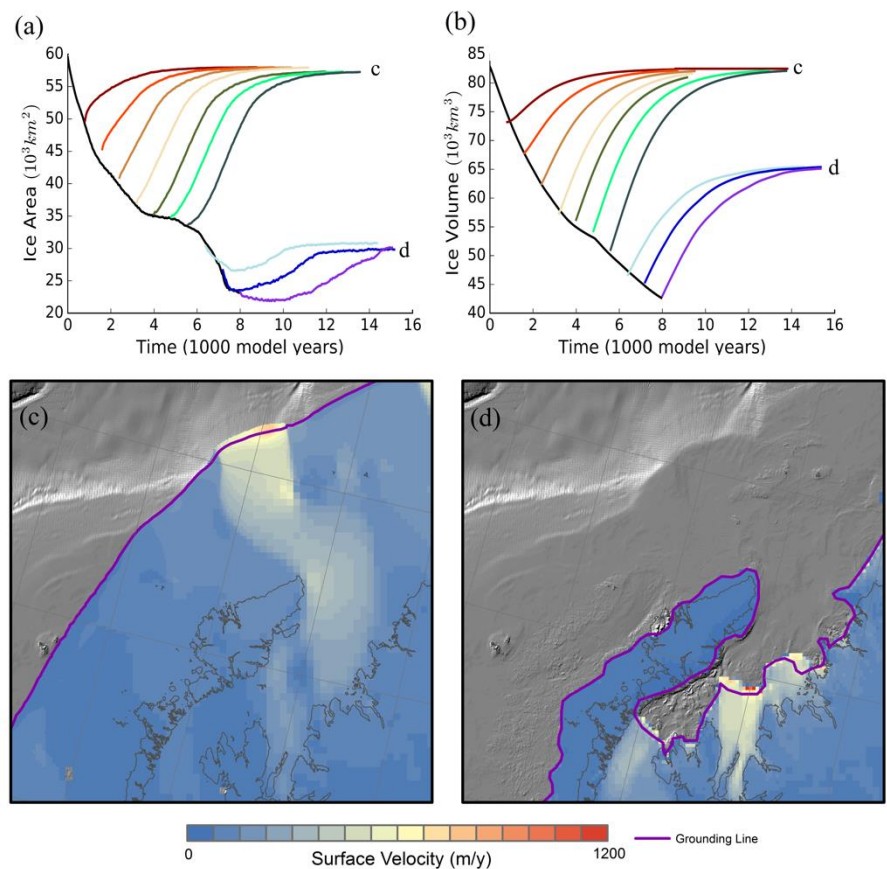

**Figure 7: Results of the ensemble of *READVANCE* simulations testing instability in the Minch Ice Stream. (a, b) Evolution of the ice sheet area (a) and volume (b) over the Minch sector in the *RETREAT* (black) and *READVANCE* simulations (coloured lines). Labels c and d indicate the "maximum" (c) and "collapsed" (d) stable states with corresponding panels showing the respective surface ice sheet velocity (m/y) and grounding line locations (purple line).**

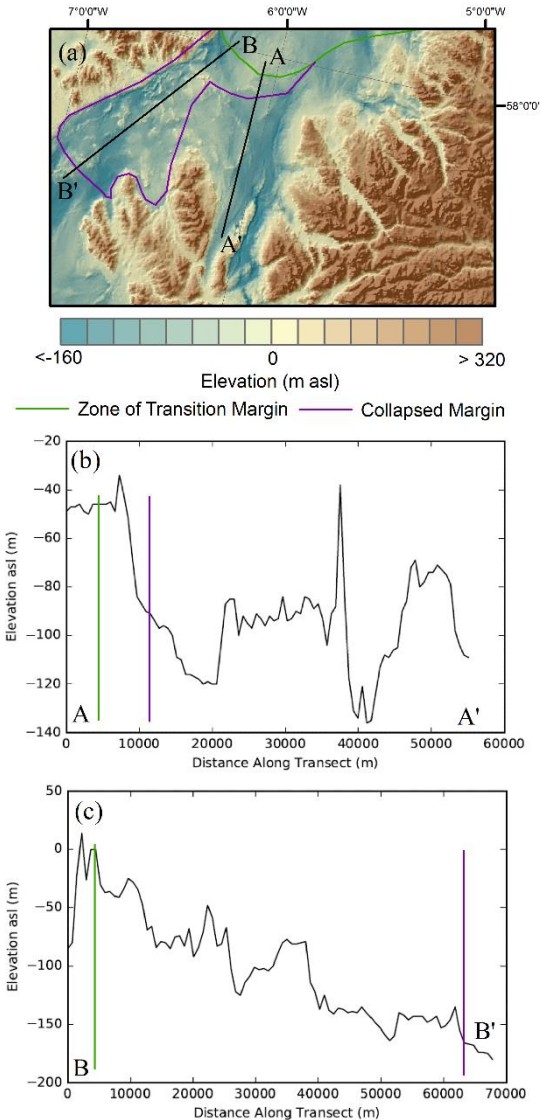

**Figure 8: The difference in the effects of MISI between the East Trough and the West Trough. a) 6,400 model year *RETREAT* margin (green) and the stable "collapsed" extent margin (purple). b) A-A' transect of bathymetry and collapse extent in the East Trough. c) B-B' transect of bathymetry and collapse extent in the West Trough.**