# Peer review of "Marine Ice Sheet Instability and Ice Shelf Buttressing Influenced Deglaciation of the Minch Ice Stream, Northwest Scotland"

_The Cryosphere, 2018_

## Referee Comment (RC1) · J. Seguinot (Referee) · 28 Aug 2018

N. Gandy et al. present an application of the higher-order ice sheet model BISICLES to early deglaciation dynamics of a marine-based sector of the British-Irish Ice Sheet in northwest Scotland. The Minch, a submarine trough located between mainland Scotland and the Isle of Lewis, is presented as one of the best documented marine sectors of the former ice sheet, much suited for this exercise.

The numerical experiment is divided in three stages. First, a "spin-up" run is initiated upon a previously published perfect-plastic ice geometry and brings the model to a steady state. Second, "retreat" simulations triggered by an instant increase in air temperature and sub-shelf melt are used to analyse deglaciation patterns and sensitivity to ice-shelf buttressing. Third, "readvance" simulations started at different stages of deglaciation with instant return to "spin-up" conditions are used to investigate the reversibility of marine ice "retreat". The simulations evidence the potential for ice retreat in two stages separated by a phase of thinning but near stagnation of the ice margin, and a point-of-no-return after which deglaciation becomes irreversible.

Strongly simplifying assumptions are made (and acknowledged), but the novelty of this study lies in its regional and marine focus. In fact, the authors should be credited for making one of very few (if any other) attempts to date to use the rich, newly available submarine glacial geomorphologic record to validate marine ice sheet modelling, a topic full of physical uncertainties and numerical difficulties. The manuscript is very well written and clearly illustrated.

I strongly support publication of these results, but I would encourage the authors to increase the transparency of their methods and clear a few inconsistencies in the interpretation before publication.

**1   General comments**

**Code availability and reproducibility**

As of now, the description of the methods lacks important detail and parameter values (see also specific comments below), which hinders the reproducibility and the traceability of the model results. Since a detailed sensitivity is not (and probably need not) be included, this means that readers have to trust the authors for having made reasonable model choices and used error-proof tools. This is not reasonable given the multiple uncertainties that affect ice sheet modelling.

The statement that "Cornford et al. (2013) provides a full description of BISICLES"

(p. 3, l. 20–21) is not entirely correct as that paper primarily describes the numerical treatment of ice flow and grounding line migration but not boundary model components, and the code has presumably evolved since 2013. Ice sheet models such as BISICLES are complex programs containing numerous uncertain ice physical parameters, multiple numerical approximations and configurable regularizations schemes. Most importantly, they are not exempt from coding errors.

I am certainly not advocating to clutter the article with a full model description here, which would be both disruptive and inefficient. Instead, I think that model code should be made available and the version used clearly stated in the manuscript. The required section on "code and data availability" is missing. I also suggest that authors include a referenced list of the most important ice-physical parameters used in their set-up.

**Supplementary material**

Related to the previous point, parts of the methods, and results from two sensitivity tests are given in the supplementary. I don't really understand the authors' choice to store this important information in a supplementary file (using a proprietary format with no guarantee of long-term readability). I found it a bit difficult to follow the main text without looking at supplementary figures.

I suggest that Table S1, Figs. S1 and S2, and the description of the basal friction map (with added values and for the friction coefficient, and possibly a reference to the basal sliding equation) are incorporated in the main text. Perhaps Fig. S1 could be merged with Fig. 3, and Fig. S2 with Fig. 4. References are already in place where needed. Fig. S3 does not add much to Fig. 7 and could probably be omitted.

**Fast retreat and irreversible retreat**

Some of the sentences in the abstract and conclusions (see specific comments) appear to amalgamate rapid retreat and irreversibility. However, it is worth noticing that

the area of slow retreat at the straight's mouth (best visible in the supplementary animation) is distinct from the point of irreversibility in shallow waters further upstream. This interesting result indicates a different role of lateral and vertical confinement on marginal retreat. I find this result particularly interesting in the context of the variety of currently observed patterns of tidewater glacier retreat in Greenland and Antarctica, and believe that it should be emphasized in the abstract and conclusions.

**2 Specific comments**

**p. 1, l. 17**: the ice stream becomes laterally confined at a "pinning-point"

The phrase "pinning-point" is often used to described ice rises and other contact points stabilizing an ice shelve, which may be confusing. Inline with my general comment above, I suggest replacing with a more descriptive term, e.g. the "straight's mouth" or "the end of the trough".

**p. 1, l. 18**: the presence of ice shelves became a major control on deglaciation

This statement does not really reflect the model results and argumentations in the main text (also see other comment below). Actually I was surprised to find how little effect ice shelves have on the pace of deglaciation, for which authors provide a very satisfactory explanation. I think this sentence should be reworked.

**p. 2, l. 1–3**: The previous generation of ice sheet models do not accurately simulate the position of the grounding line, due to the use of the Shallow Ice Approximation (van der Veen, 2013).

It would make sense to mention possible subgrid parametrization of shallow approximations here (Feldmann et al., 2014).

**p. 3, l. 21–22**: we set up the model domain to cover the majority of the British-Irish Ice Sheet

No mention is made of model results outside of the Minch domain. Does the model performs reasonably well there too?

**p. 4, l. 3–4**: we initialise ice thickness in the domain from a perfectly plastic ice sheet model

How were other variables, especially ice temperature, initialized?

**p. 4, l. 5–6**: the 27ka BP margin of Clark et al. (2012) was used for the remainder of the BIIS.

Was the 27 ka BP margin used as a boundary for the plastic model? If so, in which sense is it fortunate that the plastic ice sheet matches it well? If not, how was ice extent converted to ice thickness?

**p. 4, l. 8–10**: Positive Degree Day (PDD) mass balance model, as described by Seguinot (2013) and Gregoire et al. (2016, 2015).

None of the given references contain a full PDD model description. Actually Gregoire et al. (2016) do not even mention PDD models. Does the model resolves the sub-annual evolution of snow cover? Does it account for meltwater refreezing? Most importantly, which PDD factors are used? Does the reference to (Seguinot, 2013) imply that the authors incorporate daily temperature variability from HadCAM3, or perhaps use the code I wrote for this publication (https://github.com/juseg/pypdd)? Or is the PDD model part of BISICLES? I think a new paragraph is needed here to address these questions.

**p. 4, l. 18**: To remove the effect of a SMB feedback

This is an important simplification. Although it may be justifiable under Antarctic-like settings, in a warmer climate the surface elevation-mass balance feedback could strongly affect the pace of margin retreat and hysteresis effects discussed further in the manuscript. In this context, introducing a few details about input climate would help. I wonder which parts of the model domain are affected by seasonal melt, and to which extent dynamic thinning due to ice surface lowering would affect the the afore-mentioned conclusions.

**p. 4, l. 23**: sub-shelf melt (m/y) is 10× the annual average SST (K)

Melt rate and temperature are distinct physical quantities. For mathematical rigour I suggest to rework the sentence and introduce an equation and a unit to the factor 10.

**p. 4, l. 27–28**: We correct the SST to permit for ice shelf formation

Presumably both modelled SST and the aforementioned factor 10 suffer from uncertainties. However, what is the justification for applying a correction on input data rather than a model parameter? Does this correction replaces or supplement the model bias correction mentioned earlier in the manuscript?

**p. 4, l. 32**: studies of present day ice streams using the same friction law

On which modern ice streams were these studies conducted?

**p. 5, l. 2–3**: the extent of ice after a 6,000 year spin-up remains comparable even with different magnitudes of basal friction.

Is this because the bedrock topography drops abruptly at the continental shelf edge? Thus changes in basal friction affect ice thickness but not its extent?

Besides, Fig. S1 shows a significant drop in ice volume (and presumably ice thickness)

for all runs. Does this decrease concerns the entire BIIS domain, or only the marine-influenced sector of the Minch?

**p. 5, l. 6**: Isostatic adjustment was simulated using the EUST3 GIA model

I think that a short model description and list of parameters used is needed here. Is the GIA model ran at equilibrium or transiently? Is the value of mantle viscosity below the Minch well constrained?

**p. 5, l. 14**: Experiment set-up is summarised in the supplement (Table S1).

I think this table should be part of the main text, and supplemented by a second table listing other, constant model parameters (see general comment above).

**p. 5, l. 24**: We keep bed topography and sea level constant

Similarly to the assumption on surface elevation-mass balance feedback, I think that this second assumption needs to be put in context of the regional bedrock properties. Is the bedrock response expected to be instantaneous or delayed? What is the expected rate of relative sea-level drop and how could this affect the model results?

**Fig. 4 y-label**: Ice volume ($10^4 \, \mathrm{km}^2$)

Dividing volume by area yields an ice thickness of ca. 20 km. There must be an order-of-magnitude error here.

**p. 6, l. 17**: halving the domain's ice volume in the first 2,000 model years

This contradicts Fig. 4 where ice volume appears to drop from ca. 83 to ca. 62 axes units, corresponding to only around a quarter of the initial ice thickness.

**p. 7, l. 21**: the volume change caused by removing an ice shelf is significant.

In the context of earlier simplifications on surface mass balance and bedrock topography, I am not convinced that a 10% change of volume is significant. I suspect that a few sensitivity tests on input climate, basal sliding parameters and uncertain bedrock properties would yield much larger changes in ice volumes (cf. e.g., Seguinot et al., 2016). I would simply remove this statement, and correct the corresponding sentence in the abstract (see previous comment).

**p. 8, l. 15–16**: the ice stream does not recover to ILGM extent [...] (Figure 7a,b)

I suggest to refer to Fig. 7 here.

**p. 8, l. 16–17**: a volume 25% smaller, and an area 50% smaller [...] state (Figure 7d).

I think Fig. 7a and b would be more appropriate.

**p. 8, l. 18–19**: a full shelf-edge glaciation

One could refer to Fig. 7c here...

**p. 8, l. 19**: a small Hebrides Ice Cap with glaciation in the Minch limited to the east trough (Figure 7)

And Fig. 7d here.

**p. 8, l. 21–23**: Hysteresis of ice sheet evolution is evidence for an instability during the advance of retreat of an ice sheet (Schoof, 2007). Ice sheets can experience a variety of instabilities (Calov et al., 2002; Gregoire et al., 2016, 2012; Schoof, 2007) which could influence the ice sheet evolution.

I do not really understand how these two sentences relate to the surrounding discussion. Without formal analysis of all intermediate stable states it would be more correct to write that the model results hint at an hysteresis (formally identified by Schoof, 2007). Studies by (Gregoire et al., 2012, 2016) concern a surface elevation-mass balance ice-sheet instability which is not only unrelated but precisely missing in the current study, so I would just remove the second sentence, or move it to a discussion of potential weaknesses.

**p. 9, l. 5–6**: east of the trough [...] west of the trough

If I am not mistaken, on Fig. 1 these are labelled "east trough" and "west trough".

**p. 9, l. 1–3**: the east trough contains a small ice stream whilst the west trough has fully deglaciated and formed a calving bay.

A reference to geological observations is needed here.

**p. 9, l. 32–33**: Due to the idealised climate forcing, only the pattern, and not the timing, of the retreat can be compared to empirical reconstructions.

After the Mynch has been announced as well-documented by geology, and since the main model results concerns the (at least relative) timing of deglaciation, this sentence comes very disenchanting! I think it could be reworked to something more positive.

**p. 10, l. 9–10**: Here, we define the reconstructed retreat [...] as the "observed retreat".

I think this definition is oversimplifying and potentially misleading. The discussion could become more constructive if it instead made clear what are geomorphological evidence, what are sedimentological evidence, and what are geological reconstructions containing a part of interpretation.

**p. 10, l. 19**: GZW6

I assume this means "Grounding zone wedge 6". Could this be added somewhere on
a map?

**p. 11, l. 3**: However, the observed retreat began at 30ka BP, when the NGRIP $\delta18O$
record suggests a cooling climate globally

The connection between numerous arguments against interpreting Greenland $\delta18O$
records as proxies for global climate.

**p. 11, l. 8–10**: There are two likely possibilities that could explain the MnIS retreating in
a cooling climate (explained below); i) internal mechanisms caused by the expansion
of the rest of the BIIS, or ii) a local SMB change of this sector of the ice sheet at 30ka
BP.

I think that a third possibility should be considered here. The assumption of initial
steady-state during maximum extent does not necessarily hold true. Although the au-
thors discuss ice piracy from neighbouring ice sheet sectors, the possibility that the
maximum extent of Minch ice stream is itself the result of a thermodynamic destabili-
sation (a surge) following build-up of stiff colder ice during the advance phase exists.
This explanation should probably be included in the list.

**p. 12, l. 13**: This result is evidence for a rapid retreat of the MnIS caused by marine ice
sheet instability.

This sentence confuses rapid retreat with instability (see general comment above).

Finally, I would like to congratulate the authors again for their effort to bring palaeo-
glaciologic data-model comparison at sea, and hope they will find my comments useful
in bringing their manuscript to final form.

**References**

Feldmann, J., Albrecht, T., Khroulev, C., Pattyn, F., and Levermann, A.: Resolution-dependent performance of grounding line motion in a shallow model compared with a full-Stokes model according to the MISMIP3d intercomparison, Journal of Glaciology, 60, 353–360, doi: 10.3189/2014jog13j093, 2014.

Gregoire, L. J., Payne, A. J., and Valdes, P. J.: Deglacial rapid sea level rises caused by ice-sheet saddle collapses, Nature, 487, 219–222, doi:10.1038/nature11257, 2012.

Gregoire, L. J., Otto-Bliesner, B., Valdes, P. J., and Ivanovic, R.: Abrupt Bølling warming and ice saddle collapse contributions to the Meltwater Pulse 1a rapid sea level rise, Geophysical Research Letters, 43, 9130–9137, doi:10.1002/2016gl070356, 2016.

Schoof, C.: Ice sheet grounding line dynamics: Steady states, stability, and hysteresis, Journal of Geophysical Research, 112, doi:10.1029/2006jf000664, 2007.

Seguinot, J.: Spatial and seasonal effects of temperature variability in a positive degree-day glacier surface mass-balance model, J. Glaciol., 59, 1202–1204, doi: 10.3189/2013JoG13J081, 2013.

Seguinot, J., Rogozhina, I., Stroeven, A. P., Margold, M., and Kleman, J.: Numerical simulations of the Cordilleran ice sheet through the last glacial cycle, The Cryosphere, 10, 639–664, doi: 10.5194/tc-10-639-2016, 2016.

---

## Referee Comment (RC2) · Anonymous Referee #2 · 7 Sep 2018

This manuscript describes a series of simulations of the deglaciation of the Minch Ice Stream (MnIS) region within the British-Irish Ice Sheet. Using BISICLES, the authors determine that a region of reverse-sloping bed causes rapid deglaciation of the MnIS through the marine ice sheet instability. They also determine that MnIS begins in an unconfined state during the LGM wherein the development of an ice shelf has little influence on the ice stream, but then transitions to a confined state as the ice stream retreats into a region of pinning.

Overall, the paper is well written, besides a few awkward and confusing explanations (pointed out below). The simulations that are conducted are clever and effectively

show that the marine ice sheet instability and (maybe) buttressing may play a role in deglaciation of MnIS. I think the most general issues that need to be addressed in this study are: (1) To what extent are these simulations supposed to represent actual deglaciation vs. numerical experiments on the role of some ice dynamical feedbacks in this region? (2) Can you be sure that some of the other feedbacks which have been omitted purposefully (to focus on MISI/buttressing alone) do not play a role in modifying the importance of MISI/buttressing? (3) How important is buttressing, actually?

I think that what is needed to address these issues is either: (a) providing a stronger argument why these processes can be excluded and the conclusions of the study remain intact, (b) additional simulations which explore the role of these other feedbacks and buttressing more carefully. I think that is these issues are addressed (in addition to the more minor issues listed below), this paper would be a valuable contribution to understanding the deglaciation of the British-Irish Ice Sheet and ice sheet dynamics more broadly.

Major points: 1. I think it needs to be made clear that the uncertainties associated with the climate and glaciology of the MnIS during deglaciations are large enough that the simulations presented here are not necessarily representative of the actual time-dependent deglaciation, but rather investigate mechanisms that may have played a role during a generic deglaciation of MnIS. That is, you make the following assumptions in the RETREAT simulation (that I suppose is the most like actual deglaciation): climate forcing was step-like, there was an ice shelf (i.e. calving and basal melt were low enough to allow for ice shelf formation), basal friction was fixed in time, and SMB does not evolve with changing elevation. You say that you make these simplifications so that you can isolate internal instabilities. This point is important to make more clearly and upfront as the purpose of this study, since making these simplifications takes you away somewhat from reality.

2. You have wisely decided to perform some of your numerical simulations while omitting certain feedbacks, in order to test whether MISI and buttressing may occur given

certain bed topography and climate forcing. This does not answer the question of whether MISI/buttressing are the most important feedbacks, or whether they might be significantly amplified or reduced by cooperating feedbacks. In particular: rebound of the bed and the elevation-SMB feedback may also play an important role in the deglaciation of ice streams. Gomez et al (Nature Geo, 2010) have shown that local sea level changes may have an influence on grounding line stability during deglaciation. Robel & Tziperman (JGR, 2016) have shown that curvature of the elevation-SMB feedback may cause acceleration of ice stream during the initial stages of deglaciation. It may be worth addressing these issues by making an argument for why these other feedbacks are not important, and also potentially performing additional simulations that test the influence of these feedbacks to some extent.

3. Page 4, Line 22 (and page 5, line 12 and page 10, line 7-9): The elevation-SMB interaction may be considered to be an "internal instability" considering that it mostly has to do with the climate at the ice sheet surface. Another way to think about this, why do we only care about "internal" instabilities? Are these the only instabilities that are important? To what extent can we disentangle feedbacks having to do with the ice sheet from those having to do with the ice sheet surface climate?

4. Page 5, Line 26: I can see that "the purpose of this study is to test for a retreat instability of the MnIS with a given topography" However, why is this the correct purpose for the study to have? What does this tell us about the actual deglaciation or about the process of MISI? I think you need to go a bit further than stating that the purpose is to do some clever simulations to the purpose is to answer some particular science question.

5. How important is buttressing, actually? (Page 7, Line 18-20) You've said that the influence of the ice shelf on the grounding line position is very small, so how much influence does the ice shelf actually have on grounding line stability? If you include floating ice in your calculation of ice volume, then it is not clear to me how much of your 10% volume difference is just the ice shelf itself. I think you need to be a little more

careful in order to make a strong argument that there is any buttressing actually happening here. Especially since this is a much wider ice stream than what you typically find in Greenland (see argument p.7, line 32), it is not obvious that the buttressing will be significant.

Minor Points: page 1 Line 11: "A valuable case to examine these processes is - awkward phrasing Line 13: what is well constrained? the ice stream or a measurement about the ice stream? Line 15: continental shelf Line 16: sub-ice shelf melt Line 21: We conclude that geological data...the future of contemporary ice sheets. Line 29: Consequently, any change in ice thickness at the grounding line can cause an irreversible grounding line migration with no change in external forcing. [The point is that the term "instability in grounding line migration" is unclear.]

page 2 Line 1: which can stabilize Line 3: Higher-order models have more success in accurately simulation the grounding line... Line 5: Cite Tsai et al. 2015, JGlac Line 8: episodic retreat Line 8-9: how is retreat controlled by retreat history? Line 12: proxy observations (no hyphen)

page 3 Line 17: with L1L2 physics retained from the full Stokes flow equations (Schoof and Hindmarsh 2010). Line 18: BISCICLES uses adaptive... Line 26: Is there reasons to think that a linear Weertman exponent represents ice stream dynamics effectively? Is there a good citation? Line 28-30: This would be a good place to explain first that you pick these values specifically in order to produce large ice shelves. Otherwise, it just seems like you just replace calving with a constant parameter (not much of a "calving model").

page 4 Line 2: We set the simulations initial conditions to the ice sheet state when the MnIS was at its... Line 5: The 27 ka BP margin...of the BIIS, which matches well with the reconstructed... [In general, you don't need to use the phase "for the purpose of this study", which doesn't convey any information] Line 14: Do the climate model simulations used to force your model take into account ice sheet topography?

[Figure]

If so, say here. If not, you need to justify why this is reasonable. Lines 23-28: If you are eventually just going to strongly adjust the sub-shelf basal melt rate to retain ice shelves, why introduce the linear parameterization in the first place? Why not just simplify this whole part by setting some arbitrary sub-shelf melt rate that either permits or removes ice shelves?

page 5 Line 1: in what sense do you mean magnitude here? Line 13: I'm a bit confused. . .you initialize with a plastic thickness approximation, but experiments are then begun with a stable lLGM volume. Does the SPINUP run get you from the plastic thickness initial condition to the stable lLGM configuration? The initialization proce- dure should be explained a bit more clearly. Line 17: How were the magnitudes of the climate perturbations chosen? Are they backed up by modeling evidence for the deglacial change in climate here?

page 6 Line 14-15: Why would you expect the rate of volume and area change to be constant throughout the simulation? If anything, the initial phase of retreat looks something like exponential decay, which is approximately what one would expect from a fairly simple linear response model (i.e. dV/dt = -a*V). Classical work on the ice sheet response time scale under forcing (Nye 1960, 1963, 1965; Jóhannesson et al., Ââ1989; Harrison et al., 2003) finds such an exponential decay response.
Line 27: As the thinning continues, the ice area begins to retreat. . .

page 7 Line 2: period after hindmarsh citation. Line 3: Ice stream acceleration in response to the sudden collapse. . . Line 14: how much of the 10% difference in ice volume in the NOSHELF simulation is accounted for by the volume of the ice shelf itself (assuming this isn't volume above flotation, and if so, you should indicate that). Line 20: If you turned the calving rate way down and got a much larger ice shelf, could you stabilize on the retrograde slope?

page 8 Line 26-31: These lines include a lot of redundant information. Could be cleaned up. Line 32: marine or shear margin?

page 10 Line 19-20: You start talking about GZW6 suddenly here. Bears more explanation and/or pointing to figures. Line 34: It is also the case that a change in ice stream velocity might cause a significant change in ice sheet volume (i.e. through acceleration-induced thinning), but not much change in area. This has a lot to do with local bed topography.

page 11 Line 4-6: As mentioned above, it may be the case that triggering of ice stream acceleration by appropriately-structured climate forcing may cause a larger retreat (see Robel & Tziperman 2016).

page 12 Line 1: provide citations for why hydrology may or may not be important

Figure 3, Panel b: colormap could use more constant, very difficult to see any difference Figures 3, 4, 5, 6, 7: The axis labels and tick labels are far too small and illegible. Also the lines in all plots could have greater thickness to increase legibility. It also looks like the text is grainy because the resolution of the figures is low. Please include higher resolution figures.

---

## Author Comment (AC1) · 18 Oct 2018

N. Gandy et al. present an application of the higher-order ice sheet model BISICLES to early deglaciation dynamics of a marine-based sector of the British-Irish Ice Sheet in northwest Scotland. The Minch, a submarine trough located between mainland Scotland and the Isle of Lewis, is presented as one of the best documented marine sectors of the former ice sheet, much suited for this exercise.

The numerical experiment is divided in three stages. First, a "spin-up" run is initiated upon a previously published perfect-plastic ice geometry and brings the model to a steady state. Second, "retreat" simulations triggered by an instant increase in air temperature and sub-shelf melt are used to analyse deglaciation patterns and sensitivity to ice-shelf buttressing. Third, "readvance" simulations started at different stages of deglaciation with instant return to "spin-up" conditions are used to investigate the reversibility of marine ice "retreat". The simulations evidence the potential for ice retreat in two stages separated by a phase of thinning but near stagnation of the ice margin, and a point-of-no-return after which deglaciation becomes irreversible.

Strongly simplifying assumptions are made (and acknowledged), but the novelty of this study lies in its regional and marine focus. In fact, the authors should be credited for making one of very few (if any other) attempts to date to use the rich, newly available submarine glacial geomorphologic record to validate marine ice sheet modelling, a topic full of physical uncertainties and numerical difficulties. The manuscript is very well written and clearly illustrated.

I strongly support publication of these results, but I would encourage the authors to increase the transparency of their methods and clear a few inconsistencies in the interpretation before publication.

**1 General comments**

*Code availability and reproducibility*

As of now, the description of the methods lacks important detail and parameter values (see also specific comments below), which hinders the reproducibility and the traceability of the model results. Since a detailed sensitivity is not (and probably need not) be included, this means that readers have to trust the authors for having made reasonable model choices and used error-proof tools. This is not reasonable given the multiple uncertainties that affect ice sheet modelling.

The statement that "Cornford et al. (2013) provides a full description of BISICLES" (p. 3, l. 20–21) is not entirely correct as that paper primarily describes the numerical treatment of ice flow and grounding line migration but not boundary model components, and the code has presumably evolved since 2013. Ice sheet models such as BISICLES are complex programs containing numerous uncertain ice physical parameters, multiple numerical approximations and configurable regularizations schemes. Most importantly, they are not exempt from coding errors.

I am certainly not advocating to clutter the article with a full model description here, which would be both disruptive and inefficient. Instead, I think that model code should be made available and the version used clearly stated in the manuscript. The required section on "code and data availability" is

missing. I also suggest that authors include a referenced list of the most important ice-physical parameters used in their set-up.

We have now included the required code and data availability section. It provides a link to the version of the BISICLES model code used. It also includes a DOI for the deposited data described in the manuscript. This comprises all input data needed to run the experiments, plus our outputs for each experiment. A link to the PyPDD model is also included.

A list of the most important model parameters is now included in table 1.

*Supplementary material*

Related to the previous point, parts of the methods, and results from two sensitivity tests are given in the supplementary. I don't really understand the authors' choice to store this important information in a supplementary file (using a proprietary format with no guarantee of long-term readability). I found it a bit difficult to follow the main text without looking at supplementary figures.

I suggest that Table S1, Figs. S1 and S2, and the description of the basal friction map (with added values and for the friction coefficient, and possibly a reference to the basal sliding equation) are incorporated in the main text. Perhaps Fig. S1 could be merged with Fig. 3, and Fig. S2 with Fig. 4. References are already in place where needed. Fig. S3 does not add much to Fig. 7 and could probably be omitted.

To make the manuscript easier to follow we have moved Table S1, and Figure S1 and S2 to the main text as suggested. A description of the basal friction map is now included in section 3.2 (Initial conditions and spin-up). Figure S3 has been removed from the Supplementary Information, as recommended.

*Fast retreat and irreversible retreat*

Some of the sentences in the abstract and conclusions (see specific comments) appear to amalgamate rapid retreat and irreversibility. However, it is worth noticing that the area of slow retreat at the strait's mouth (best visible in the supplementary animation) is distinct from the point of irreversibility in shallow waters further upstream. This interesting result indicates a different role of lateral and vertical confinement on marginal retreat. I find this result particularly interesting in the context of the variety of currently observed patterns of tidewater glacier retreat in Greenland and Antarctica, and believe that it should be emphasized in the abstract and conclusions.

We have edited the text to separate the discussion of fast and irreversible retreat from the abstract and conclusion. We have also added sentences (p1,l21)(p8,l7)(p14,l18) to mention the important comparison between later stages of retreat and tidewater glaciers of Greenland and Antarctica.

**2 Specific comments**

p. 1, l. 17: the ice stream becomes laterally confined at a "pinning-point"
The phrase "pinning-point" is often used to described ice rises and other contact points stabilizing an ice shelve, which may be confusing. Inline with my general comment above, I suggest replacing with a more descriptive term, e.g. the "straight's mouth" or "the end of the trough"

The phrase "pinning-point" has been changed to "the Minch Strait's mouth" throughout the manuscript to avoid potential confusion.

p. 1, l. 18: the presence of ice shelves became a major control on deglaciation

This statement does not really reflect the model results and argumentations in the main text (also see other comment below). Actually I was surprised to find how little effect ice shelves have on the pace of deglaciation, for which authors provide a very satisfactory explanation. I think this sentence should be reworked.

The sentence has been reworded to better reflect the model results (p1,l21);

"At this stage of the simulation, the presence of an ice shelf becomes a more important control on grounded ice volume"

p. 2, l. 1–3: The previous generation of ice sheet models do not accurately simulate the position of the grounding line, due to the use of the Shallow Ice Approximation (van der Veen, 2013).

It would make sense to mention possible subgrid parametrization of shallow approximations here (Feldmann et al., 2014).

Done (p2,l8)

p. 3, l. 21–22: we set up the model domain to cover the majority of the British-Irish Ice Sheet

No mention is made of model results outside of the Minch domain. Does the model performs reasonably well there too?

The experiments described in this manuscript were set up specifically to model the MnIS. Deglaciation is generally poorly represented outside the Minch catchment for a handful of reasons. Primarily, proximity to the domain edge causes over-stability of the ice sheet along the eastern margin. The bed friction map (figure 2c) is also likely missing a number of ice stream paths, the map is potentially most complete in the northwest of the domain. The model resolution is also much coarser outside the Minch catchment (4 km at the grounding line rather than 500 m). Overall the Barra Fan Ice Streams catchment (Hughes et al., 2014) (https://doi.org/10.1016/j.quascirev.2014.02.002) is the closest to also being well represented (it is located far from a domain edge and is in the bed friction map), but is still modelled at only 4 km resolution. We have not referenced model results outside the Minch domain to keep the focus on the single catchment and resulting experimental design.

p. 4, l. 3–4: we initialise ice thickness in the domain from a perfectly plastic ice sheet model

How were other variables, especially ice temperature, initialized?

We have included a new table (Table 1) that documents key model parameters, including ice temperature.

p. 4, l. 5–6: the 27ka BP margin of Clark et al. (2012) was used for the remainder of the BIIS.

Was the 27 ka BP margin used as a boundary for the plastic model? If so, in which sense is it fortunate that the plastic ice sheet matches it well? If not, how was ice extent converted to ice thickness?

Yes, the 27 ka BP margin was used in the plastic model (figure 2a). The match between the 27 ka margin the Bradwell et al., (2008) maximum extent was described as fortunate because 27 ka BP is not the global LGM, nor the maximum extent of the BIIS, but it does reasonably represent MnIS maximum extent. However, the word fortunate is removed as it doesn't convey any useful meaning.

p. 4, l. 8–10: Positive Degree Day (PDD) mass balance model, as described by Seguinot (2013) and Gregoire et al. (2016, 2015).

None of the given references contain a full PDD model description. Actually Gregoire et al. (2016) do not even mention PDD models. Does the model resolves the subannual evolution of snow cover? Does it account for meltwater refreezing? Most importantly, which PDD factors are used? Does the reference to (Seguinot, 2013) imply that the authors incorporate daily temperature variability from HadCAM3, or perhaps use the code I wrote for this publication (https://github.com/juseg/pypdd)? Or is the PDD model part of BISICLES? I think a new paragraph is needed here to address these questions.

We used the PyPDD script here and have amended the text to specify this. A more explicit reference to the code is now made, and referenced to Seguinot (2013) as requested on the PyPDD GitHub page. The PDD factors, refreeze ratio, and snow/rain threshold is now summarized in table 1.

5 Gregoire et al. (2016, 2015) references examples of using palaeo-gcm results to drive ice sheet SMB. We have clarified this in the text.

p. 4, l. 18: To remove the effect of a SMB feedback

10 This is an important simplification. Although it may be justifiable under Antarctic like settings, in a warmer climate the surface elevation-mass balance feedback could strongly affect the pace of margin retreat and hysteresis effects discussed further in the manuscript. In this context, introducing a few details about input climate would help. I wonder which parts of the model domain are affected by seasonal melt, and to which extent dynamic thinning due to ice surface lowering would affect the the
15 aforementioned conclusions.

The potential for interaction between MISI and SMB-elevation feedback (or other simplifications made) was also mentioned by reviewer #2 (please see below). We have edited the manuscript to better highlight and justify the simplifications made, primarily in the form of an extended section 4.4
20 (Comparison to empirical reconstructions).

p. 4, l. 23: sub-shelf melt (m/y) is 10× the annual average SST (K)

Melt rate and temperature are distinct physical quantities. For mathematical rigour I suggest to
25 rework the sentence and introduce an equation and a unit to the factor 10.

Done (p5,l4)

p. 4, l. 27–28: We correct the SST to permit for ice shelf formation
30
Presumably both modelled SST and the aforementioned factor 10 suffer from uncertainties. However, what is the justification for applying a correction on input data rather than a model parameter? Does this correction replaces or supplement the model bias correction mentioned earlier in the manuscript?

35 Both the modelled SST and sub-shelf melt/SST relationship do indeed have very large uncertainties. In fact, Holland et al., (2008) (https://doi.org/10.1175/2007JCLI1909.1) compared 9 relationships between ice shelf melting and ocean temperature. The Rignot and Jacobs (2002) rate we use is broadly representative of these relationships, but there is a large spread. We corrected the input data instead of the parameter because the modelled SST also has large uncertainties, and is a simple linear

correction. Maintaining the linear parameterization allows for easier comparison to Antarctic and Greenland ice shelves. The sub-shelf melt rates are supplementary to the frontal ablation rates discussed in section 3.1 (Model description and set-up).

5   Please note that the reference to Iyengar et al. (2001) was an error caused by the referencing software used, and has been manually corrected to Rignot and Jacobs (2002).

p. 4, l. 32: studies of present day ice streams using the same friction law

10  On which modern ice streams were these studies conducted?

Pine Island Glacier, and Basin 3 (Austfonna). Clarification added (p6,l3).

p. 5, l. 2–3: the extent of ice after a 6,000 year spin-up remains comparable even with different
15  magnitudes of basal friction.

Is this because the bedrock topography drops abruptly at the continental shelf edge? Thus changes in basal friction affect ice thickness but not its extent? Besides, Fig. S1 shows a significant drop in ice volume (and presumably ice thickness) for all runs. Does this decrease concerns the entire BIIS
20  domain, or only the marine influenced sector of the Minch?

For the Minch catchment the extent of the ice sheet is strongly controlled by the position of the shelf edge, the increase in water depth is sudden and dramatic. However, in other sectors of the ice sheet the extent is also controlled by the SMB map (Figure 2). We have now highlighted this in the text
25  (p6,l7).

p. 5, l. 6: Isostatic adjustment was simulated using the EUST3 GIA model

I think that a short model description and list of parameters used is needed here. Is the GIA model ran
30  at equilibrium or transiently? Is the value of mantle viscosity below the Minch well constrained?

The GIA model simulations with the best fit to sea level index points used a thin lithosphere (71 km), an upper mantle viscosity of 4-6 x$10^{20}$ Pa s, and a lower mantle viscosity ≥ 3 x$10^{20}$ Pa s. The model was run transiently by Bradley et al., (2011) and we used their results as input for the topographic
35  adjustment. The text has been edited to clarify this (p6,l10).

p. 5, l. 14: Experiment set-up is summarised in the supplement (Table S1).

I think this table should be part of the main text, and supplemented by a second table listing other, constant model parameters (see general comment above).

Done

p. 5, l. 24: We keep bed topography and sea level constant

Similarly to the assumption on surface elevation-mass balance feedback, I think that this second assumption needs to be put in context of the regional bedrock properties. Is the bedrock response
10 expected to be instantaneous or delayed? What is the expected rate of relative sea-level drop and how could this affect the model results?

The relative sea level (RSL) change is expected to be small, around 15 m sea level fall, causing a minimal change in marine/terrestrial areal extent. This is because of the counteracting effects of the
15 expanding BIIS (depressing the bedrock) and a concurrent drop in global sea level due to increased terrestrial glaciation. This has been placed into context in the text (p7,l6), along with an expanded section 4.4 (Comparison to empirical reconstructions) justifying the simplifying assumptions more clearly.

20 Fig. 4 y-label: Ice volume (104 km2 )

Dividing volume by area yields an ice thickness of ca. 20 km. There must be an orderof-magnitude error here.

25 Yes, a conversion error was occurring in a script used in Figures 3, 4, 5, and 7. All plots have been updated accordingly.

p. 6, l. 17: halving the domain's ice volume in the first 2,000 model years

30 This contradicts Fig. 4 where ice volume appears to drop from ca. 83 to ca. 62 axes units, corresponding to only around a quarter of the initial ice thickness.

This has been corrected in the text (p8,l3).

35 p. 7, l. 21: the volume change caused by removing an ice shelf is significant.

In the context of earlier simplifications on surface mass balance and bedrock topography, I am not convinced that a 10% change of volume is significant. I suspect that a few sensitivity tests on input climate, basal sliding parameters and uncertain bedrock properties would yield much larger changes in

ice volumes (cf. e.g., Seguinot et al., 2016). I would simply remove this statement, and correct the corresponding sentence in the abstract (see previous comment).

We have corrected the corresponding sentence in the abstract and have removed this sentence as advised.

p. 8, l. 15–16: the ice stream does not recover to lLGM extent [...] (Figure 7a,b)

I suggest to refer to Fig. 7 here.

Done (p9,l32)

p. 8, l. 16–17: a volume 25% smaller, and an area 50% smaller [...] state (Figure 7d).

I think Fig. 7a and b would be more appropriate.

Done (p9,l33)

p. 8, l. 18–19: a full shelf-edge glaciation

One could refer to Fig. 7c here...

Done (p10,l2)

p. 8, l. 19: a small Hebrides Ice Cap with glaciation in the Minch limited to the east trough (Figure 7)

And Fig. 7d here.

Done (p10,l2)

p. 8, l. 21–23: Hysteresis of ice sheet evolution is evidence for an instability during the advance of retreat of an ice sheet (Schoof, 2007). Ice sheets can experience a variety of instabilities (Calov et al., 2002; Gregoire et al., 2016, 2012; Schoof, 2007) which could influence the ice sheet evolution.

I do not really understand how these two sentences relate to the surrounding discussion. Without formal analysis of all intermediate stable states it would be more correct to write that the model results hint at an hysteresis (formally identified by Schoof, 2007). Studies by (Gregoire et al., 2012, 2016) concern a surface elevation-mass balance icesheet instability which is not only unrelated but

precisely missing in the current study, so I would just remove the second sentence, or move it to a discussion of potential weaknesses.

The second sentence has been removed as advised, the interaction between MISI and other ignored factors is now discussed in more length in section 4.4 (Comparison to empirical reconstructions). The wording of the first sentence has been altered (p10,l4).

p. 9, l. 5–6: east of the trough [...] west of the trough

If I am not mistaken, on Fig. 1 these are labelled "east trough" and "west trough".

We have corrected this to give consistent names between map and reference in the text (p10,l22).

p. 9, l. 1–3: the east trough contains a small ice stream whilst the west trough has fully deglaciated and formed a calving bay.

A reference to geological observations is needed here.

Reference added (p11,l31).

p. 9, l. 32–33: Due to the idealised climate forcing, only the pattern, and not the timing, of the retreat can be compared to empirical reconstructions.

After the Mynch has been announced as well-documented by geology, and since the main model results concerns the (at least relative) timing of deglaciation, this sentence comes very disenchanting! I think it could be reworked to something more positive.

We think it is important to highlight that the nature of the forcing means the modelled retreat timing cannot be compared to the empirical data, but the style and comparative rate of retreat can. The dating evidence here is strong and growing, but it should be compared to these model results only considering these caveats. The sentence has been reworded to change the focus (p11,l27).

p. 10, l. 9–10: Here, we define the reconstructed retreat [...] as the "observed retreat".

I think this definition is oversimplifying and potentially misleading. The discussion could become more constructive if it instead made clear what are geomorphological evidence, what are sedimentological evidence, and what are geological reconstructions containing a part of interpretation.

The definition of observed retreat was an attempt to make the comparisons between the modelled data and the empirically reconstructed data more clear. The definition has been removed, and references to the observed retreat have been replaced with comparisons to "empirical reconstructions", including in the section header.

p. 10, l. 19: GZW6

I assume this means "Grounding zone wedge 6". Could this be added somewhere on a map?

GZW6 mistakenly refers to an unpublished map (Bradwell et al., in review). This reference has been removed from the text. References to grounding zone wedges have been replaced with "large seabed moraines", as they have been referred to in previous literature (Clark et al., 2018; Stoker and Bradwell, 2015) (p12,l11).

p. 11, l. 3: However, the observed retreat began at 30ka BP, when the NGRIP δ18O record suggests a cooling climate globally

The connection between numerous arguments against interpreting Greenland δ18O records as proxies for global climate.

This has been corrected to suggest only a cooling climate of Greenland (p13,l1). Model results instead suggest a generally cooling climate, and have been referenced in the text (p13,l2).

p. 11, l. 8–10: There are two likely possibilities that could explain the MnIS retreating in a cooling climate (explained below); i) internal mechanisms caused by the expansion of the rest of the BIIS, or ii) a local SMB change of this sector of the ice sheet at 30ka BP.

I think that a third possibility should be considered here. The assumption of initial steady-state during maximum extent does not necessarily hold true. Although the authors discuss ice piracy from neighbouring ice sheet sectors, the possibility that the maximum extent of Minch ice stream is itself the result of a thermodynamic destabilisation (a surge) following build-up of stiff colder ice during the advance phase exists. This explanation should probably be included in the list.

The possibility of thermodynamic destabilization as a possible internal mechanism has been added to the discussion (p13,l26). This is an important point. Dating the advance of an ice sheet is inherently more uncertain than dating retreat, so it is perfectly possible given the empirical reconstructions that the advance of the MnIS was a rapid surge type event.

p. 12, l. 13: This result is evidence for a rapid retreat of the MnIS caused by marine ice sheet instability.

This sentence confuses rapid retreat with instability (see general comment above).

The reference to rapid retreat has been removed.

Finally, I would like to congratulate the authors again for their effort to bring palaeoglaciologic data-model comparison at sea, and hope they will find my comments useful in bringing their manuscript to final form.

**Review #2**
This manuscript describes a series of simulations of the deglaciation of the Minch Ice Stream (MnIS) region within the British-Irish Ice Sheet. Using BISICLES, the authors determine that a region of reverse-sloping bed causes rapid deglaciation of the MnIS through the marine ice sheet instability. They also determine that MnIS begins in an unconfined state during the LGM wherein the development of an ice shelf has little influence on the ice stream, but then transitions to a confined state as the ice stream retreats into a region of pinning.

Overall, the paper is well written, besides a few awkward and confusing explanations (pointed out below). The simulations that are conducted are clever and effectively show that the marine ice sheet instability and (maybe) buttressing may play a role in deglaciation of MnIS. I think the most general issues that need to be addressed in this study are: (1) To what extent are these simulations supposed to represent actual deglaciation vs. numerical experiments on the role of some ice dynamical feedbacks in this region? (2) Can you be sure that some of the other feedbacks which have been omitted purposefully (to focus on MISI/buttressing alone) do not play a role in modifying the importance of MISI/buttressing? (3) How important is buttressing, actually?

I think that what is needed to address these issues is either: (a) providing a stronger argument why these processes can be excluded and the conclusions of the study remain intact, (b) additional simulations which explore the role of these other feedbacks and buttressing more carefully. I think that is these issues are addressed (in addition to the more minor issues listed below), this paper would

be a valuable contribution to understanding the deglaciation of the British-Irish Ice Sheet and ice sheet dynamics more broadly.

**Major Points**

1. I think it needs to be made clear that the uncertainties associated with the climate and glaciology of the MnIS during deglaciations are large enough that the simulations presented here are not necessarily representative of the actual timedependent deglaciation, but rather investigate mechanisms that may have played a role during a generic deglaciation of MnIS. That is, you make the

10 following assumptions in the RETREAT simulation (that I suppose is the most like actual deglaciation): climate forcing was step-like, there was an ice shelf (i.e. calving and basal melt were low enough to allow for ice shelf formation), basal friction was fixed in time, and SMB does not evolve with changing elevation. You say that you make these simplifications so that you can isolate internal instabilities. This point is important to make more clearly and upfront as the purpose of this study, since making these

15 simplifications takes you away somewhat from reality

These simplifying assumptions are now made more explicitly in the abstract, and the first sentence of the methods sections. An increased initial discussion of the caveats in section 4.4 (Comparison to empirical reconstructions) allows for the purpose and effect of the simplifications to be put more

20 clearly and extensively in the manuscript. Further discussion on the effects of the simplifications, and the likely differences between the simulations and reality remain in section 4.4 (Comparison to observed retreat).

The purpose of the simplifying assumptions is to test if the bathymetric set-up of the Minch means

25 that the ice stream could have been vulnerable to marine influence. In reality, the numerous other factors idealised here could exaggerate or dampen the impact of MISI, but our experiments neatly demonstrate that MISI is a possible factor given the bathymetry of the Minch. This is important because hitherto, the marine nature of the BIIS has only been assumed because the ice sheet was marine grounded in significant catchments, like the Minch, and it was not known whether MISI was

30 active here (for example, the bathymetry is so considerably less extreme than in west Antarctica that it was not obvious that conditions were right for MISI). These simulations are required to demonstrate the potential effect of MISI on the BIIS, and therefore the potential role of MISI in less extreme conditions than have so far been demonstrated.

35 The original intended purpose of the experiments, to test the possibility of marine ice sheet dynamics mechanisms for the MnIS, is best preserved with this now extended discussion of the justification and effects of simplifications. To truly comment on the relative importance of MISI, for the MnIS would instead require a transient simulation of evolving bed friction, hydrology, sea level, climate, and

thermodynamics. This would represent a major and challenging project in its own right and is beyond the scope of this study.

2. You have wisely decided to perform some of your numerical simulations while omitting certain feedbacks, in order to test whether MISI and buttressing may occur given certain bed topography and climate forcing. This does not answer the question of whether MISI/buttressing are the most important feedbacks, or whether they might be significantly amplified or reduced by cooperating feedbacks. In particular: rebound of the bed and the elevation-SMB feedback may also play an important role in the deglaciation of ice streams. Gomez et al (Nature Geo, 2010) have shown that local sea level changes may have an influence on grounding line stability during deglaciation. Robel & Tziperman (JGR, 2016) have shown that curvature of the elevation-SMB feedback may cause acceleration of ice stream during the initial stages of deglaciation. It may be worth addressing these issues by making an argument for why these other feedbacks are not important, and also potentially performing additional simulations that test the influence of these feedbacks to some extent.

As stated above, we do not make an argument for these other factors not being important. In reality, these factors may have been very important. We have extended the discussion (p11,ln13) to highlight the influence and interaction of other factors. The manuscript changes (section 3.3, section 4.4) aim to make clearer that the purpose of the simplifying assumptions is to test if the bathymetric set-up of the Minch means that the ice stream could have been vulnerable to MISI, not to determine the relative importance of MISI. We have chosen to clarify the purpose and effects of the simplifying assumptions rather than perform additional simulations, as explained in our response to the previous point and also partly because it is not clear that additional simulations could truly reveal the relative importance of MISI with the current tools available.

3. Page 4, Line 22 (and page 5, line 12 and page 10, line 7-9): The elevation-SMB interaction may be considered to be an "internal instability" considering that it mostly has to do with the climate at the ice sheet surface. Another way to think about this, why do we only care about "internal" instabilities? Are these the only instabilities that are important? To what extent can we disentangle feedbacks having to do with the ice sheet from those having to do with the ice sheet surface climate?

This is an important point that primarily has been tackled in the expanded section 4.4 (Comparison to empirical reconstructions). We agree with the reviewer's implication that the difference between internal and external instabilities doesn't provide much meaning, so reference to it has been removed from the manuscript. The process of disentangling various functions of ice sheet change is also now discussed in more length in section 4.4 (Comparison to empirical reconstructions).

4. Page 5, Line 26: I can see that "the purpose of this study is to test for a retreat instability of the MnIS with a given topography" However, why is this the correct purpose for the study to have? What

does this tell us about the actual deglaciation or about the process of MISI? I think you need to go a bit further than stating that the purpose is to do some clever simulations to the purpose is to answer some particular science question.

This line has been replaced with a clearer description of the motivation for a constant sea level in the experiments (p7,l6). One purpose of the study was to understand if MISI was a possible mechanism given the bathymetric set-up of the Minch because this could not yet be otherwise determined without our new numerical experiments. The motivation for this approach is now discussed at more length in section 4.4 (Comparison to empirical reconstructions).

5. How important is buttressing, actually? (Page 7, Line 18-20) You've said that the influence of the ice shelf on the grounding line position is very small, so how much influence does the ice shelf actually have on grounding line stability? If you include floating ice in your calculation of ice volume, then it is not clear to me how much of your 10% volume difference is just the ice shelf itself. I think you need to be a little more careful in order to make a strong argument that there is any buttressing actually happening here. Especially since this is a much wider ice stream than what you typically find in Greenland (see argument p.7, line 32), it is not obvious that the buttressing will be significant.

The cited 10% volume divergence between RETREAT and RETREAT_NOSHELF is for grounded ice. This has been clarified in section 4.2 (Role of ice shelf buttressing), and in Figure 5 description. A 10% difference is important, but perhaps not as large as could be expected. What we find particularly interesting is the relationship between the ice shelf and the topographic setting, discussed in section 4.2 (Role of ice shelf buttressing).

**Minor Points**

*page 1*
Line 11: "A valuable case to examine these processes is - awkward phrasing

The order of the sentence has been changed to read more clearly (p1,l12).

Line 13: what is well constrained? the ice stream or a measurement about the ice stream?

Both – but here we meant that there is rich empirical data for the MnIS. The sentence has been reworded to avoid confusion with being topographically constrained (p1,l14).

Line 15: continental shelf

Done (p1,l18)

Line 16: sub-ice shelf melt

Changed – and changed for all other occurrences in the manuscript.

Line 21: We conclude that geological data. . .the future of contemporary ice sheets.

Done (p1,l27)

Line 29: Consequently, any change in ice thickness at the grounding line can cause an irreversible grounding line migration with no change in external forcing. [The point is that the term "instability in grounding line migration" is unclear.]

Done (p2,l3)

*page 2*
Line 1: which can stabilize

Done (p2,l5)

Line 3: Higher-order models have more success in accurately simulation the grounding line

Done (p2,l9)

Line 5: Cite Tsai et al. 2015, JGlac

Done (p2,l11)

Line 8: episodic retreat

Done (p2,l15)

Line 8-9: how is retreat controlled by retreat history?

The delayed upstream response of ice streams means that long-term projections of ice sheets "should carefully integrate long-term ice-stream history" (Jamieson et al., 2012). We think the important point here is upstream response, and the text has been updated accordingly (p2,l16).

Line 12: proxy observations (no hyphen)

Done (p2,l19)

*page 3*

Line 17: with L1L2 physics retained from the full Stokes flow equations (Schoof and Hindmarsh 2010).

Done (p3,l25)

Line 18: BISICLES uses adaptive. . .

Done (p3,l27)

Line 26: Is there reasons to think that a linear Weertman exponent represents ice stream dynamics effectively? Is there a good citation?

A linear Weertman exponent was used for other ice stream modelling studies using BISICLES (Favier et al., 2014; Gong et al., 2017), from which the bed friction coefficient map values were based. This has now been added to the manuscript (p4,l4). The ice stream velocity is primarily controlled by the prescribed bed friction map (figure 2c). Using an alternative Weertman exponent would alter the velocity magnitude and volume of the ice stream, as was the case for variations in the bed friction coefficient (figure 3a).

Line 28-30: This would be a good place to explain first that you pick these values specifically in order to produce large ice shelves. Otherwise, it just seems like you just replace calving with a constant parameter (not much of a "calving model").

Done (p4,l9)

*page 4*

Line 2: We set the simulations initial conditions to the ice sheet state when the MnIS was at its. . .

Done (p4,l12)

Line 5: The 27 ka BP margin. . .of the BIIS, which matches well with the reconstructed. . . [In general, you don't need to use the phase "for the purpose of this study", which doesn't convey any information]

"for the purpose of this study" has been removed.

Line 14: Do the climate model simulations used to force your model take into account ice sheet topography? If so, say here. If not, you need to justify why this is reasonable.

Ice sheet orography was updated in the simulations used here since the simulations reported by Singarayer et al., (2011). This has been included in the text (p4,l26). More information on the climate simulations used as input has been included in this section of the manuscript.

Lines 23-28: If you are eventually just going to strongly adjust the sub-shelf basal melt rate to retain ice shelves, why introduce the linear parameterization in the first place? Why not just simplify this whole part by setting some arbitrary sub-shelf melt rate that either permits or removes ice shelves?

The linear parameterization is used because it aids interpretation of the results by putting them into the context of current ice streams in Greenland and Antarctica. Although setting the values arbitrarily is possible, it could be mechanistically different to what is observed. It therefore remains best practice to base the values (as we have done) on observed melt rates in order to preserve the mechanisms of reality.

**page 5**
Line 1: in what sense do you mean magnitude here?

Magnitude was the incorrect term to use here, replaced with morphology (p6,l4)

Line 13: I'm a bit confused. . .you initialize with a plastic thickness approximation, but experiments are then begun with a stable lLGM volume. Does the SPINUP run get you from the plastic thickness initial condition to the stable lLGM configuration? The initialization procedure should be explained a bit more clearly.

Table 2 now shows a summary of experimental design without the reader needing to refer to the supplementary information. A new first sentence added to section 3.3.1 (Deglaciation) also clarifies how ice thickness is initialized for the experiment.

We initialize ice thickness of the *SPIN-UP* experiment with the plastic thickness approximation, which then evolves during the simulation (Figure 3c). The end of the *SPIN-UP* experiment is used as the start of all other experiments (apart from *READVANCE* experiments).

Line 17: How were the magnitudes of the climate perturbations chosen? Are they backed up by modeling evidence for the deglacial change in climate here?

The magnitudes of change were based on the magnitude of change between 26 and 18 ka BP from the equilibrium climate modelling data we used. This has been clarified in the text (p6,l28).

**page 6**

5   Line 14-15: Why would you expect the rate of volume and area change to be constant throughout the simulation? If anything, the initial phase of retreat looks something like exponential decay, which is approximately what one would expect from a fairly simple linear response model (i.e. dV/dt = -a*V). Classical work on the ice sheet response time scale under forcing (Nye 1960, 1963, 1965; Jóhannesson et al., Âa1989; Harrison et al., 2003) finds such an exponential decay response.â ˘ A˘´I

We have changed the wording to instead focus on the fluctuation in area and volume response, and to properly acknowledge prior work (as suggested by the reviewer) that indicates an exponential decay in ice volume would be expected (p7,l30).

15   Line 27: As the thinning continues, the ice area begins to retreat. . .

Done (p8,l14)

**page 7**

20   Line 2: period after hindmarsh citation.

Done (p8,l17)

Line 3: Ice stream acceleration in response to the sudden collapse. . .

Done (p8,l19)

Line 14: how much of the 10% difference in ice volume in the NOSHELF simulation is accounted for by the volume of the ice shelf itself (assuming this isn't volume above flotation, and if so, you should

30   indicate that).

The 10% difference in ice volume is for the grounded ice volumes of *RETREAT* and *RETREAT_NOSHELF*. This has now been clarified in section 4.2 (Role of ice shelf buttressing), and in the figure 5 heading.

35   Line 20: If you turned the calving rate way down and got a much larger ice shelf, could you stabilize on the retrograde slope?

Running the deglaciation without an increase in the calving rate does not prompt full deglaciation (figure 4a). A decreased calving rate therefore cannot stabilize the groundling line on a retrograde

slope, because without the increased calving rate the grounding line does not reach the retrograde slope.

**page 8**

Line 26-31: These lines include a lot of redundant information. Could be cleaned up.

A significant amount of repetition has been removed here.

Line 32: marine or shear margin?

Marine – clarified in the text. (p10,l16)

**page 10**

Line 19-20: You start talking about GZW6 suddenly here. Bears more explanation and/or pointing to figures.

Please see the response to a comment from reviewer #1;

GZW6 mistakenly refers to an unpublished map (Bradwell et al., in review). This reference has been removed from the text.

Line 34: It is also the case that a change in ice stream velocity might cause a significant change in ice sheet volume (i.e. through acceleration-induced thinning), but not much change in area. This has a lot to do with local bed topography.

We have added this important example of the effect of ice stream dynamics to the text (p12,l29). The importance of ice stream velocity is evident from the variations in bed friction tested during the *SPIN-UP* experiment, which produced different ice volumes but the same ice area, controlled by the position of the continental shelf.

**page 11**

Line 4-6: As mentioned above, it may be the case that triggering of ice stream acceleration by appropriately-structured climate forcing may cause a larger retreat (see Robel & Tziperman 2016).

This mechanism is indeed a candidate to prompt rapid deglaciation of the MnIS, though an initial climate trigger is required to increase the surface slopes at the ice sheet margin. An acknowledgement of the mechanism has been added to the text (p13,l3).

**page 12**

Line 1: provide citations for why hydrology may or may not be important

Done (p14,l8)

Figure 3, Panel b: colormap could use more constant, very difficult to see any difference

Done

Figures 3, 4, 5, 6, 7: The axis labels and tick labels are far too small and illegible. Also the lines in all plots could have greater thickness to increase legibility. It also looks like the text is grainy because the resolution of the figures is low. Please include higher resolution figures.

The axis labels, tick labels, and plot curves have all been increased in size. All plots are now saved at 1000dpi.

[revised manuscript text omitted]
 Ttable 2. the supplement (Table S1). is(Bradley et al., 2011; Clark et al., 2012; Scourse et al., 2018)These experiments are designed to test the applicability of MISI to the MnIS.

20 In reality, ice sheet evolution is a function of climate fluctuations, ice surface evolution, and sea level change as well as MISI. However, to isolate the effects of MISI, sea level is held constant throughout the experiments, there is no elevation-SMB feedback, and the climate change is a simple step-change.

in the experimental design were made to investigate the mechanism of in the MnIS, and as such, approach [LG1][DH2]

**3.3.1 Deglaciation**

[revised manuscript text omitted]

---

## Editor Decision (ED1)

[revised manuscript text omitted]
 Ttable 2. the supplement (Table S1). is(Bradley et al., 2011; Clark et al., 2012; Scourse et al., 2018)These experiments are designed to test the applicability of MISI to the MnIS.

In reality, ice sheet evolution is a function of climate fluctuations, ice surface evolution, and sea level change as well as MISI. However, to isolate the effects of MISI, sea level is held constant throughout the experiments, there is no elevation-SMB feedback, and the climate change is a simple step-change.

In the experimental design were made to investigate the mechanism of in the MnIS, and as such , approach [LG1][DH2]

**3.3.1 Deglaciation**

[revised manuscript text omitted]

T Verfasser: oeisen    Thema: Hervorheben Datum: 28.10.18, 19:37:47
Figure 6: b) Please order line labes according to years! Currently it looks like a mess, more difficult in this way to understand.

As mentioned above, it would be advantageous if the MTBH would be marked in the figure, e.g. as a small bar at the bottom. At least explained with km in the caption.

Verfasser: oeisen     Thema: Hervorheben Datum: 28.10.18, 19:28:13
add reference to where the colour code is explained, i.e. Fig. 6.

[Figure]

**Figure 7: Results of the ensemble of *READVANCE* simulations testing instability in the Minch Ice Stream. (a, b) Evolution of the ice sheet area (a) and volume (b) over the Minch sector in the *RETREAT* (black) and *READVANCE* simulations (coloured lines). Labels c and d indicate the "maximum" (c) and "collapsed" (d) stable states with corresponding panels showing the respective surface ice sheet velocity (m/y) and grounding line locations (purple line).**

**Figure 8: The difference in the effects of MISI between the East Trough and the West Trough. a) 6,400 model year *RETREAT* margin (green) and the stable "collapsed" extent margin (purple). b) A-A' transect of bathymetry and collapse extent in the East Trough. c) B-B' transect of bathymetry and collapse extent in the West Trough.**

---

## Author Response (AR2)

**Response to Editor: minor revisions**

We are pleased to resubmit this manuscript after recommended minor revisions. We trust we have adequately addressed the points raised by the editor, and that the revised manuscript is improved as a result. We would like to thank the editor for their helpful and prompt comments.

Editor comments are in black

Author responses are in blue

**Editor comments**

p. 2, l. 14: Marguerite Bay Ice Stream, Antarctica,

Done (p. 2, l. 10)

p. 4, l. 10: Figures should appear in the order they are referenced in the text.

A prior reference to Figure 3 has now been made (p. 4, l. 7). We made this earlier reference to Figure 3 to allow the order of figures to be preserved, as Figure 4 (The evolution of the ice sheet during the *RETREAT* experiment) is best understood after Figure 3 (The evolution of the ice sheet during *SPIN-UP*).

p. 4, l. 12: at

Done (p. 4, l. 10)

p. 5, l. 5: .

Done (p. 5, l. 1)

p. 5, l. 6: W

Done (p. 5, l. 2)

p. 5, l. 22: coefficients (β); 100.

Sufficient to have beta explained in parentheses once (basal friction coefficient (β)), no need to repeat that in every item

p. 5, l. 27: (β=2,000).

Done (p. 5, l. 20)

p. 5, l. 30: β=3,000 was prescribed here

Done (p. 5, l. 23)

p. 5, l. 33: (β=4,000).

Done (p. 5, l. 26)

p. 6, l.5: See above: figures should be sorted in the order of there referencing in the text.

A prior reference to Figure 3 ensures figures are cited in the text in the correct order (see above).

p. 8, l. 15: (Figure

Done (p. 8, l. 6)

p. 9, l. 26: ad position in km, as not indicated in Fig. 6.

Alternatively, mark in Fig. 6b.

The position of the MTBH is now marked in Figure 6b.

p. 11, l. 21: beyond the capability of current

Done (p. 11, l. 6)

p. 11, l. 30: East Trough

Done (p. 11, l. 14)

p. 11, l. 30: West Trough

Done (p. 11, l. 14)

p. 11, l. 31: This is a reference to the reconstructed retreat. Please add a reference to a Fig, of the simulated retreat, I assume that 7d?

The structure of the sentence has been altered slightly to accommodate a reference to both empirical and simulation evidence;

"In the later stages of retreat the East Trough contains a small ice stream whilst the West Trough has fully deglaciated and formed a calving bay, evident both in the simulated (Figure 7d), and reconstructed retreat (Bradwell and Stoker, 2015)."

p. 11, l. 32: you use reconstructed retreat/simulated retreat above and now simulated deglaciation and empirical deglaciation reconstructions.

First, keep the order the same, second, unify by only using one set of words. If you stick to both, it might be useful to shortly explain somewhere in the MS that you mean the same. Otherwise it looks more complicated that it is in fact.

Simulated retreat and reconstructed retreat are now referred to consistently (p. 11 l. 15)(p. 11 l. 17)(p. 11 l. 23)(p. 12 l. 13).

p. 11, l. 33: do you think it is really "replicated", which means the processes are correct? Wouldn't mimicking be more appropriate?

"Mimicked" is a more appropriate choice, and this has been changed (p. 11 l. 18).

p. 12, l. 27: ,

Done (p. 12, l. 7)

p. 12, l. 34: clarify if you mean simulated retreat

We are referring to the empirically reconstructed retreat, and have clarified this in the text (p. 12, l. 13).

p. 14, l. 1: 2016). Although

Done (p. 13, l. 14)

p. 20: Doesn't fit in the table. Delete here, sufficient to mention in text.

This has been removed from the table.

p. 21: Would be useful to add some words on the columns initial ice thickness and surface mass balance in the caption directly so that it is self-explained.

Figures showing the initial ice thickness and surface mass balance (Figure 2a-b) are now highlighted in the table caption.

p. 23, l. 2: Add "Contemporary coast shown as thin black line" or alike in caption.

Also applies to Figs 4, 5.

Added for Figs 2, 3, 4, 5, and 7.

p. 23, l. 3: Is this water or ice equ.?

Water equivalent. This is now clarified in the caption.

p. 23, l. 4: (

Done (p. 23, l. 4)

p. 23, l. 4: bed

Done (p. 23, l. 5)

p. 24: y-axis unit not italic in (a)

also Fig 4&5.

Y-axis units are no longer in italics for Figures 3, 4, 5, and 7.

p. 25, l. 4: ,

Done (p. 25, l. 5)

p. 25, l. 4: ,

Done (p. 25, l. 5)

p. 27: Figure 6: b) Please order line labels according to years! Currently is looks like a mess, more difficult in this way to understand.

As mentioned above, it would be advantageous if the MTBG would be marked in the figure, e.g. as a small bar at the bottom.

At least explain with km in the caption.

The line labels have now been ordered correctly. A grey bar now marks the position of the MTBH along the transect.

p. 28: add reference to where the colour code is explained, i.e. Fig. 6.

The timing of the margin/transect plotting is slightly different between Figure 6 and 7 (every 800 years, starting at 1,200 and 800 years respectively). In the previous manuscript the Figure 7 curves were coloured arbitrary. They have now been coloured to correspond to the prior margin position in Figure 6. The caption has been updated accordingly (p. 28, l. 4).

[revised manuscript text omitted]

**Figure 1: Location and geographical setting of the Minch Ice Stream. The inset shows the maximum extent of the British-Irish Ice Sheet from Clark et al. (2012). The red box indicates the Minch ice stream region shown in the larger map. Ice margins and dates are from Clark et al. (2012). Moraines are from the BRITICE glacial landform map, version 2 (Clark et al., 2018). Area of potential**
5  **MISI vulnerability is inferred from the presence of marine retrograde slopes. Key locations mentioned in the text are labelled.**

[Figure]

**Figure 2: Initial conditions and boundary conditions for the equilibrium spin-up lLGM ice sheet simulation, showing the full domain of the simulations. Contemporary coast is shown as a thin grey line. (a) Initial ice thickness (m), (b) Annual Surface Mass Balance (m/y), with black contour line representing the Equilibrium Line (SMB = 0 m w.e./y). (c) Bed Friction Coefficient (β), (d) Isostatically adjusted bed topography (m) corresponding to 30 ka BP. The maps show the full ice sheet domain and the black boxes indicate the area of model grid refinement referred to as the Minch sector.**

[Figure]

**Figure 3: a) Evolution of the ice sheet volume over the Minch sector during spin-up. Contemporary coast is shown as a thin grey line. b) Simulated ice sheet thickness after 6,000 model years of spin-up, c) Ice thickness change between the start and end (year 6000) of the spin-up simulation (i.e. difference between Figure 2a and Figure 3b.)**

[Figure]

**Figure 4: Evolution of the ice sheet in the Minch sector in the *RETREAT* simulation. Contemporary coast is shown as a thin grey line. a) Timeseries of ice volume and area over the Minch sector. The dashed curve shows the volume response to *RETREAT_OCN*. The dotted curve shows the response to *RETREAT_ATMOS*. Ice surface velocity (b, c, d, e, and f) at 0, 2000, 3500, 6300, and 7500**
5 **model years, respectively, with grounding line shown in purple.**

[Figure]

**Figure 5: Effect of ice shelves.** Contemporary coast is shown as a thin grey line. **(a) Evolution of grounded ice sheet volume over the Minch sector in simulation** *RETREAT* **with ice shelves (red line) and** *RETREAT_NOSHELF* **in which ice shelves are forcibly removed (blue line). (b,c) Surface velocity and grounding line location (purple line) in RETREAT_NOSHELF (b) and** *RETREAT* **(c).**

[Figure]

**Figure 6: a) The grounding line position of the ice sheet at 800 year intervals during the *RETREAT* simulation. b) Transect through the ice stream showing the ice sheet elevation at the same intervals. The purple line marks the grounding line position in the stable "collapsed" ice sheet state from the *READVANCE* simulations (Figure 7d).**

[Figure]

**Figure 7: Results of the ensemble of *READVANCE* simulations testing instability in the Minch Ice Stream. Contemporary coast is shown as a thin grey line. (a, b) Evolution of the ice sheet area (a) and volume (b) over the Minch sector in the *RETREAT* (black) and *READVANCE* simulations (coloured lines). Line colours correspond to those of Figure 6, such that the initialisation of the *READVANCE* simulations is 400 years later than the timing of the grounding line positions shown in Figure 6. Labels c and d indicate the "maximum" (c) and "collapsed" (d) stable states with corresponding panels showing the respective surface ice sheet velocity (m/y) and grounding line locations (purple line).**

[Figure]

**Figure 8: The difference in the effects of MISI between the East Trough and the West Trough. a) 6,400 model year *RETREAT* margin (green) and the stable "collapsed" extent margin (purple). b) A-A' transect of bathymetry and collapse extent in the East Trough. c) B-B' transect of bathymetry and collapse extent in the West Trough.**

---

## Author Response (AR3)

**Response to Editor: minor revisions**

We are pleased to resubmit this manuscript after recommended minor revisions. We trust we have adequately addressed the points raised by the editor, and that the revised manuscript is improved as a result. We would like to thank the editor for their helpful and prompt comments.

Editor comments are in black

Author responses are in blue

**Editor comments**

p. 2, l. 14: Marguerite Bay Ice Stream, Antarctica,

Done (p. 2, l. 10)

p. 4, l. 10: Figures should appear in the order they are referenced in the text.

A prior reference to Figure 3 has now been made (p. 4, l. 7). We made this earlier reference to Figure 3 to allow the order of figures to be preserved, as Figure 4 (The evolution of the ice sheet during the *RETREAT* experiment) is best understood after Figure 3 (The evolution of the ice sheet during *SPIN-UP*).

p. 4, l. 12: at

Done (p. 4, l. 10)

p. 5, l. 5: .

Done (p. 5, l. 1)

p. 5, l. 6: W

Done (p. 5, l. 2)

p. 5, l. 22: coefficients (β); 100.

Sufficient to have beta explained in parentheses once (basal friction coefficient (β)), no need to repeat that in every item

p. 5, l. 27: (β=2,000).

Done (p. 5, l. 20)

p. 5, l. 30: β=3,000 was prescribed here

Done (p. 5, l. 23)

p. 5, l. 33: (β=4,000).

Done (p. 5, l. 26)

p. 6, l.5: See above: figures should be sorted in the order of there referencing in the text.

A prior reference to Figure 3 ensures figures are cited in the text in the correct order (see above).

p. 8, l. 15: (Figure

Done (p. 8, l. 6)

p. 9, l. 26: ad position in km, as not indicated in Fig. 6.

Alternatively, mark in Fig. 6b.

The position of the MTBH is now marked in Figure 6b.

p. 11, l. 21: beyond the capability of current

Done (p. 11, l. 6)

p. 11, l. 30: East Trough

Done (p. 11, l. 14)

p. 11, l. 30: West Trough

Done (p. 11, l. 14)

p. 11, l. 31: This is a reference to the reconstructed retreat. Please add a reference to a Fig, of the simulated retreat, I assume that 7d?

The structure of the sentence has been altered slightly to accommodate a reference to both empirical and simulation evidence;

"In the later stages of retreat the East Trough contains a small ice stream whilst the West Trough has fully deglaciated and formed a calving bay, evident both in the simulated (Figure 7d), and reconstructed retreat (Bradwell and Stoker, 2015)."

p. 11, l. 32: you use reconstructed retreat/simulated retreat above and now simulated deglaciation and empirical deglaciation reconstructions.

First, keep the order the same, second, unify by only using one set of words. If you stick to both, it might be useful to shortly explain somewhere in the MS that you mean the same. Otherwise it looks more complicated that it is in fact.

Simulated retreat and reconstructed retreat are now referred to consistently (p. 11 l. 15)(p. 11 l. 17)(p. 11 l. 23)(p. 12 l. 13).

p. 11, l. 33: do you think it is really "replicated", which means the processes are correct? Wouldn't mimicking be more appropriate?

"Mimicked" is a more appropriate choice, and this has been changed (p. 11 l. 18).

p. 12, l. 27: ,

Done (p. 12, l. 7)

p. 12, l. 34: clarify if you mean simulated retreat

We are referring to the empirically reconstructed retreat, and have clarified this in the text (p. 12, l. 13).

p. 14, l. 1: 2016). Although

Done (p. 13, l. 14)

p. 20: Doesn't fit in the table. Delete here, sufficient to mention in text.

This has been removed from the table.

p. 21: Would be useful to add some words on the columns initial ice thickness and surface mass balance in the caption directly so that it is self-explained.

Figures showing the initial ice thickness and surface mass balance (Figure 2a-b) are now highlighted in the table caption.

p. 23, l. 2: Add "Contemporary coast shown as thin black line" or alike in caption.

Also applies to Figs 4, 5.

Added for Figs 2, 3, 4, 5, and 7.

p. 23, l. 3: Is this water or ice equ.?

Water equivalent. This is now clarified in the caption.

p. 23, l. 4: (

Done (p. 23, l. 4)

p. 23, l. 4: bed

Done (p. 23, l. 5)

p. 24: y-axis unit not italic in (a)

also Fig 4&5.

Y-axis units are no longer in italics for Figures 3, 4, 5, and 7.

p. 25, l. 4: ,

Done (p. 25, l. 5)

p. 25, l. 4: ,

Done (p. 25, l. 5)

p. 27: Figure 6: b) Please order line labels according to years! Currently is looks like a mess, more difficult in this way to understand.

As mentioned above, it would be advantageous if the MTBG would be marked in the figure, e.g. as a small bar at the bottom.

At least explain with km in the caption.

The line labels have now been ordered correctly. A grey bar now marks the position of the MTBH along the transect.

p. 28: add reference to where the colour code is explained, i.e. Fig. 6.

The timing of the margin/transect plotting is slightly different between Figure 6 and 7 (every 800 years, starting at 1,200 and 800 years respectively). In the previous manuscript the Figure 7 curves were coloured arbitrary. They have now been coloured to correspond to the prior margin position in Figure 6. The caption has been updated accordingly (p. 28, l. 4).

**Other Changes**

The GitHub link for the PyPDD model in the Code and Data Availability section has been replaced with a DOI.

[revised manuscript text omitted]
., Barnola, J. M., Bigler, M., Biscaye, P., Caillon, N., Chappellaz, J., Clausen, H. B., Dahl-Jensen, D., Fischer, H., Flückiger, J., Fritzsche, D., Fujii, Y., Goto-Azuma, K., Grønvold, K., Gundestrup, N. S., Hansson, M., Huber, C., Hvidberg, C. S., Johnsen, S. J., Jonsell, U., Jouzel, J., Kipfstuhl, S., Landais, A., Leuenberger, M., Lorrain, R., Masson-Delmotte, V., Miller, H., Motoyama, H., Narita, H., Popp, T., Rasmussen, S. O., Raynaud, D., Rothlisberger, R., Ruth, U., Samyn, D., Schwander, J., Shoji, H., Siggard-Andersen, M. L., Steffensen, J. P., Stocker, T., Sveinbjörnsdóttir, A. E.,

Svensson, A., Takata, M., Tison, J. L., Thorsteinsson, T., Watanabe, O., Wilhelms, F. and White, J. W. C.: High-resolution record of Northern Hemisphere climate extending into the last interglacial period, Nature, 431(7005), 147–151, doi:10.1038/nature02805, 2004.

Ballantyne, C. K. and Stone, J. O.: Rock-slope failure at Baosbheinn, Wester Ross, NW Scotland: age and interpretation, Scottish J. Geol., 45(2), 177–181, doi:10.1144/0036-9276/01-388, 2009.

Becker, J. J., Sandwell, D. T., Smith, W. H. F., Braud, J., Binder, B., Depner, J., Fabre, D., Factor, J., Ingalls, S., Kim, S. H., Ladner, R., Marks, K., Nelson, S., Pharaoh, A., Trimmer, R., von Rosenberg, J., Wallace, G. and Weatherall, P.: Global Bathymetry and Elevation Data at 30 Arc Seconds Resolution: SRTM30_PLUS, Mar. Geod., 32(4), 355–371, doi:10.1080/01490410903297766, 2009.

Boulton, G. and Hagdorn, M.: Glaciology of the British Isles Ice Sheet during the last glacial cycle: form, flow, streams and lobes, Quat. Sci. Rev., 25(23–24), 3359–3390, doi:10.1016/j.quascirev.2006.10.013, 2006.

Boulton, G. S., Hagdorn, M. and Hulton, N. R. J.: Streaming flow in an ice sheet through a glacial cycle, Ann. Glaciol., 36, 117–128, doi:10.3189/172756403781816293, 2003.

Bradley, S. L., Milne, G. A., Shennan, I. and Edwards, R.: An improved glacial isostatic adjustment model for the British Isles, J. Quat. Sci., 26(5), 541–552, doi:10.1002/jqs.1481, 2011.

Bradwell, T.: Identifying palaeo-ice-stream tributaries on hard beds: Mapping glacial bedforms and erosion zones in NW Scotland, Geomorphology, 201, 397–414, doi:10.1016/j.geomorph.2013.07.014, 2013.

Bradwell, T. and Stoker, M. S.: Submarine sediment and landform record of a palaeo-ice stream within the British-Irish ice sheet, Boreas, 44(2), doi:10.1111/bor.12111, 2015.

Bradwell, T., Stoker, M. and Larter, R.: Geomorphological signature and flow dynamics of The Minch palaeo-ice stream, northwest Scotland, J. Quat. Sci., 22(6), 609–617, doi:10.1002/jqs.1080, 2007.

Bradwell, T., Stoker, M. S., Golledge, N. R., Wilson, C. K., Merritt, J. W., Long, D., Everest, J. D., Hestvik, O. B., Stevenson, A. G., Hubbard, A. L., Finlayson, A. G. and Mathers, H. E.: The northern sector of the last British Ice Sheet: Maximum extent and demise, , doi:10.1016/j.earscirev.2008.01.008, 2008.

Clark, C. D., Hughes, A. L. C., Greenwood, S. L., Jordan, C. and Sejrup, H. P.: Pattern and timing of retreat of the last British-Irish Ice Sheet, Quat. Sci. Rev., 44, 112–146, doi:10.1016/j.quascirev.2010.07.019, 2012.

Clark, C. D., Ely, J. C., Greenwood, S. L., Hughes, A. L. C., Meehan, R., Barr, I. D., Bateman, M. D., Bradwell, T., Doole, J., Evans, D. J. A., Jordan, C. J., Monteys, X., Pellicer, X. M. and Sheehy, M.: BRITICE Glacial Map, version 2: a map and GIS database of glacial landforms of the last British-Irish Ice Sheet, Boreas, 47(1), 11-e8, doi:10.1111/bor.12273, 2018.

Cornford, S. L., Martin, D. F., Graves, D. T., Ranken, D. F., Le Brocq, A. M., Gladstone, R. M., Payne, A. J., Ng, E. G. and Lipscomb, W. H.: Adaptive mesh, finite volume modeling of marine ice sheets, J. Comput. Phys., 232(1), 529–549, doi:10.1016/j.jcp.2012.08.037, 2013.

Cornford, S. L., Martin, D. F., Lee, V., Payne, A. J. and Ng, E. G.: Adaptive mesh refinement versus subgrid friction interpolation in simulations of Antarctic ice dynamics, Ann. Glaciol., 57(73), 1–9, doi:10.1017/aog.2016.13, 2016.

Dunlop, P., Shannon, R., McCabe, M., Quinn, R. and Doyle, E.: Marine geophysical evidence for ice sheet extension and recession on the Malin Shelf: New evidence for the western limits of the British Irish Ice Sheet, Mar. Geol., 276(1–4), 86–99, doi:10.1016/j.margeo.2010.07.010, 2010.

Ely, J. C., Clark, C. D., Spagnolo, M., Stokes, C. R., Greenwood, S. L., Hughes, A. L. C., Dunlop, P. and Hess, D.: Do subglacial bedforms comprise a size and shape continuum?, Geomorphology, 257, 108–119, doi:10.1016/J.GEOMORPH.2016.01.001, 2016.

Farr, T. G., Rosen, P. A., Caro, E., Crippen, R., Duren, R., Hensley, S., Kobrick, M., Paller, M., Rodriguez, E., Roth, L., Seal, D., Shaffer, S., Shimada, J., Umland, J., Werner, M., Oskin, M., Burbank, D. and Alsdorf, D.: The Shuttle Radar Topography Mission, Rev. Geophys., 45(2), RG2004, doi:10.1029/2005RG000183, 2007.

Favier, L., Durand, G., Cornford, S. L., Gudmundsson, G. H., Gagliardini, O., Gillet-Chaulet, F., Zwinger, T., Payne, A. J. and Le Brocq, A. M.: Retreat of Pine Island Glacier controlled by marine ice-sheet instability, Nat. Clim. Chang., 4(2), 117–121, doi:10.1038/nclimate2094, 2014.

Feldmann, J. and Levermann, A.: Collapse of the West Antarctic Ice Sheet after local destabilization of the Amundsen Basin., Proc. Natl. Acad. Sci. U. S. A., 112(46), 14191–6, doi:10.1073/pnas.1512482112, 2015.

Feldmann, J., Albrecht, T., Khroulev, C., Pattyn, F. and Levermann, A.: Resolution-dependent performance of grounding line motion in a shallow model compared with a full-Stokes model according to the MISMIP3d intercomparison, J. Glaciol., 60(220), 353–360, doi:10.3189/2014JoG13J093, 2014.

Funder, S., Kjeldsen, K. K., Kjær, K. H. and Ó Cofaigh, C.: The Greenland Ice Sheet During the Past 300,000 Years: A Review, Dev. Quat. Sci., 15, 699–713, doi:10.1016/B978-0-444-53447-7.00050-7, 2011.

Gladstone, R. M., Warner, R. C., Galton-Fenzi, B. K., Gagliardini, O., Zwinger, T. and Greve, R.: Marine ice sheet model performance depends on basal sliding physics and sub-shelf melting, Cryosph., 11(1), 319–329, doi:10.5194/tc-11-319-2017, 2017.

Gong, Y., Zwinger, T., Cornford, S., Gladstone, R., Schäfer, M. and Moore, J. C.: Importance of basal boundary conditions in transient simulations: Case study of a surging marine-terminating glacier on Austfonna, Svalbard, J. Glaciol., 63(237), 106–117, doi:10.1017/jog.2016.121, 2017.

Gordon, C., Cooper, C., Senior, C. A., Banks, H., Gregory, J. M., Johns, T. C., Mitchell, J. F. B. and Wood, R. A.: The simulation of SST, sea ice extents and ocean heat transports in a version of the Hadley Centre coupled model without flux adjustments, Clim. Dyn., 16(2–3), 147–168, doi:10.1007/s003820050010, 2000.

Gowan, E. J., Tregoning, P., Purcell, A., Lea, J., Fransner, O. J., Noormets, R. and Dowdeswell, J. A.: ICESHEET 1.0: a program to produce paleo-ice sheet reconstructions with minimal assumptions, Geosci. Model Dev, 9, 1673–1682, doi:10.5194/gmd-9-1673-2016, 2016.

Gregoire, L. J., Valdes, P. J. and Payne, A. J.: The relative contribution of orbital forcing and greenhouse gases to the North American deglaciation, Geophys. Res. Lett., 42(22), 9970–9979, doi:10.1002/2015GL066005, 2015.

Gregoire, L. J., Otto-Bliesner, B., Valdes, P. J. and Ivanovic, R.: Abrupt Bølling warming and ice saddle collapse contributions

to the Meltwater Pulse 1a rapid sea level rise, Geophys. Res. Lett., 43(17), 9130–9137, doi:10.1002/2016GL070356, 2016.

Gudmundsson, G. H.: Ice-shelf buttressing and the stability of marine ice sheets, Cryosph., 7, 647–655, doi:10.5194/tc-7-647-2013, 2013.

Harrison, W. D., Raymond, C. F., Echelmeyer, K. A. and Krimmel, R. M.: A macroscopic approach to glacier dynamics, J. Glaciol., 49(164), 13–21, doi:10.3189/172756503781830917, 2003.

Hewitt, I. J.: Seasonal changes in ice sheet motion due to melt water lubrication, Earth Planet. Sci. Lett., 371–372, 16–25, doi:10.1016/J.EPSL.2013.04.022, 2013.

Hindmarsh, R. C. A.: The role of membrane-like stresses in determining the stability and sensitivity of the Antarctic ice sheets: back pressure and grounding line motion., Philos. Trans. A. Math. Phys. Eng. Sci., 364(1844), 1733–67, doi:10.1098/rsta.2006.1797, 2006.

Hubbard, A., Bradwell, T., Golledge, N., Hall, A., Patton, H., Sugden, D., Cooper, R. and Stoker, M.: Dynamic cycles, ice streams and their impact on the extent, chronology and deglaciation of the British–Irish ice sheet, Quat. Sci. Rev., 28(7–8), 758–776, doi:10.1016/j.quascirev.2008.12.026, 2009.

Hughes, A. L. C., Gyllencreutz, R., Lohne, Ø. S., Mangerud, J. and Svendsen, J. I.: The last Eurasian ice sheets - a chronological database and time-slice reconstruction, DATED-1, Boreas, 45(1), doi:10.1111/bor.12142, 2016.

Ivanovic, R. F., Gregoire, L. J., Kageyama, M., Roche, D. M., Valdes, P. J., Burke, A., Drummond, R., Peltier, W. R. and Tarasov, L.: Transient climate simulations of the deglaciation 21–9 thousand years before present (version 1) – PMIP4 Core experiment design and boundary conditions, Geosci. Model Dev., 9(7), 2563–2587, doi:10.5194/gmd-9-2563-2016, 2016.

Jamieson, S. S. R., Vieli, A., Livingstone, S. J., Cofaigh, C. Ó., Stokes, C., Hillenbrand, C.-D. and Dowdeswell, J. A.: Ice-stream stability on a reverse bed slope, Nat. Geosci., 5(11), 799–802, doi:10.1038/ngeo1600, 2012.

Jenkins, A., Dutrieux, P., Jacobs, S. S., McPhail, S. D., Perrett, J. R., Webb, A. T. and White, D.: Observations beneath Pine Island Glacier in West Antarctica and implications for its retreat, Nat. Geosci., 3(7), 468–472, doi:10.1038/ngeo890, 2010.

Johannesson, T., Raymond, C. F. and Waddington, E. D.: A simple method for determining the response time of glaciers, in Glacier Fluctuations and Climatic Change, pp. 343–352., 1989.

Joughin, I., Smith, B.E., & Medley, B.: Marine Ice Sheet Collapse Potentially Under Way for the Thwautes Glacier, Science (80-. )., 344(May), 735–738, doi:10.1126/science.1249055, 2014.

Joughin, I., Howat, I. M., Fahnestock, M., Smith, B., Krabill, W., Alley, R. B., Stern, H. and Truffer, M.: Continued evolution of Jakobshavn Isbrae following its rapid speedup, J. Geophys. Res., 113(F4), F04006, doi:10.1029/2008JF001023, 2008.

Margold, M., Stokes, C. R. and Clark, C. D.: Ice streams in the Laurentide Ice Sheet: Identification, characteristics and comparison to modern ice sheets, Earth-Science Rev., 143, 117–146, doi:10.1016/J.EARSCIREV.2015.01.011, 2015.

Matsuoka, K., Hindmarsh, R. C. A., Moholdt, G., Bentley, M. J., Pritchard, H. D., Brown, J., Conway, H., Drews, R., Durand, G., Goldberg, D., Hattermann, T., Kingslake, J., Lenaerts, J. T. M., Martín, C., Mulvaney, R., Nicholls, K. W., Pattyn, F., Ross, N., Scambos, T. and Whitehouse, P. L.: Antarctic ice rises and rumples: Their properties and significance for ice-sheet dynamics and evolution, Earth Sci. Rev., 150, 724–745, doi:10.1016/j.earscirev.2015.09.004, 2015.

Morris, P. J., Swindles, G. T., Valdes, P. J., Ivanovic, R. F., Gregoire, L. J., Smith, M. W., Tarasov, L., Haywood, A. M. and Bacon, K. L.: Global peatland initiation driven by regionally asynchronous warming, Proc. Natl. Acad. Sci., 201717838, doi:10.1073/pnas.1717838115, 2018.

Nias, I. J., Cornford, S. L. and Payne, A. J.: Contrasting the Modelled sensitivity of the Amundsen Sea Embayment ice streams, J. Glaciol., 62(233), 552–562, doi:10.1017/jog.2016.40, 2016.

Nye, J. F.: The Response of Glaciers and Ice-Sheets to Seasonal and Climatic Changes, Proc. R. Soc. A Math. Phys. Eng. Sci., 256(1287), 559–584, doi:10.1098/rspa.1960.0127, 1960.

Nye, J. F.: On the Theory of the Advance and Retreat of Glaciers, Geophys. J. R. Astron. Soc., 7(4), 431–456, doi:10.1111/j.1365-246X.1963.tb07087.x, 1963.

Oster, J. L., Ibarra, D. E., Winnick, M. J. and Maher, K.: Steering of westerly storms over western North America at the Last Glacial Maximum, Nat. Geosci., 8(3), 201–205, doi:10.1038/ngeo2365, 2015.

Patton, H., Hubbard, A., Andreassen, K., Winsborrow, M. and Stroeven, A. P.: The build-up, configuration, and dynamical sensitivity of the Eurasian ice-sheet complex to Late Weichselian climatic and oceanic forcing, Quat. Sci. Rev., 153, 97–121, doi:10.1016/j.quascirev.2016.10.009, 2016.

Patton, H., Hubbard, A., Andreassen, K., Auriac, A., Whitehouse, P. L., Stroeven, A. P., Shackleton, C., Winsborrow, M., Heyman, J. and Hall, A. M.: Deglaciation of the Eurasian ice sheet complex, Quat. Sci. Rev., 169, 148–172, doi:10.1016/j.quascirev.2017.05.019, 2017.

Pattyn, F., Schoof, C., Perichon, L., Hindmarsh, R. C. A., Bueler, E., De Fleurian, B., Durand, G., Gagliardini, O., Gladstone, R., Goldberg, D., Gudmundsson, G. H., Huybrechts, P., Lee, V., Nick, F. M., Payne, A. J., Pollard, D., Rybak, O., Saito, F. and Vieli, A.: Results of the Marine Ice Sheet Model Intercomparison Project, MISMIP, Cryosph., 6, 573–588, doi:10.5194/tc-6-573-2012, 2012.

Pope, V. D., Gallani, M. L., Rowntree, P. R. and Stratton, R. A.: The impact of new physical parametrizations in the Hadley Centre climate model: HadAM3, Clim. Dyn., 16(2–3), 123–146, doi:10.1007/s003820050009, 2000.

Pritchard, H. D., Ligtenberg, S. R. M., Fricker, H. A., Vaughan, D. G., van den Broeke, M. R. and Padman, L.: Antarctic ice-sheet loss driven by basal melting of ice shelves, Nature, 484(7395), 502–505, doi:10.1038/nature10968, 2012.

Rignot, E. and Jacobs, S. S.: Rapid Bottom Melting Widespread near Antarctic Ice Sheet Grounding Lines, Science (80-. )., 296(5575), 2020–2023 [online] Available from: http://science.sciencemag.org/content/296/5575/2020 (Accessed 6 June 2017), 2002.

Ritz, C., Edwards, T. L., Durand, G., Payne, A. J., Peyaud, V. and Hindmarsh, R. C. A.: Potential sea-level rise from Antarctic ice-sheet instability constrained by observations, Nature, 528(7580), 115–118, doi:10.1038/nature16147, 2015.

Robel, A. A. and Tziperman, E.: The role of ice stream dynamics in deglaciation, J. Geophys. Res. Earth Surf., 121(8), 1540–1554, doi:10.1002/2016JF003937, 2016.

Ross, N., Bingham, R. G., Corr, H. F. J., Ferraccioli, F., Jordan, T. A., Le Brocq, A., Rippin, D. M., Young, D., Blankenship, D. D. and Siegert, M. J.: Steep reverse bed slope at the grounding line of the Weddell Sea sector in West Antarctica, Nat.

Geosci., 5(6), 393–396, doi:10.1038/ngeo1468, 2012.

Scambos, T. A., Bohlander, J. A., Shuman, C. A. and Skvarca, P.: Glacier acceleration and thinning after ice shelf collapse in the Larsen B embayment, Antarctica, Geophys. Res. Lett., 31(18), L18402, doi:10.1029/2004GL020670, 2004.

Schoof, C.: Ice sheet grounding line dynamics: Steady states, stability, and hysteresis, J. Geophys. Res., 112(F3), F03S28, doi:10.1029/2006JF000664, 2007.

Schoof, C. and Hindmarsh, R. C. A.: Thin-film flows with wall slip: An asymptotic analysis of higher order glacier flow models, Q. J. Mech. Appl. Math., 63(1), 73–114, doi:10.1093/qjmam/hbp025, 2010.

Scourse, J. D., Haapaniemi, A. I., Colmenero-Hidalgo, E., Peck, V. L., Hall, I. R., Austin, W. E. N., Knutz, P. C. and Zahn, R.: Growth, dynamics and deglaciation of the last British–Irish ice sheet: the deep-sea ice-rafted detritus record, Quat. Sci. Rev., 28(27–28), 3066–3084, doi:10.1016/J.QUASCIREV.2009.08.009, 2009.

Seguinot, J.: Spatial and seasonal effects of temperature variability in a positive degree-day glacier surface mass-balance model, J. Glaciol., 59(218), 1202–1204, doi:10.3189/2013JoG13J081, 2013.

Siegert, M. J. and Dowdeswell, J. A.: Numerical reconstructions of the Eurasian Ice Sheet and climate during the Late Weichselian, Quat. Sci. Rev., 23(11–13), 1273–1283, doi:10.1016/J.QUASCIREV.2003.12.010, 2004.

Singarayer, J. S. and Valdes, P. J.: High-latitude climate sensitivity to ice-sheet forcing over the last 120 kyr, , doi:10.1016/j.quascirev.2009.10.011, 2010.

Singarayer, J. S., Valdes, P. J., Friedlingstein, P., Nelson, S. and Beerling, D. J.: Late Holocene methane rise caused by orbitally controlled increase in tropical sources, Nature, 470(7332), 82–86, doi:10.1038/nature09739, 2011.

Stoker, M. S. and Bradwell, T.: The Minch palaeo-ice stream, NW sector of the British–Irish Ice Sheet, J. Geol. Soc. London., 162(3), 425–428, doi:10.1144/0016-764904-151, 2005.

Stokes, C. R. and Clark, C. D.: Geomorphological criteria for identifying Pleistocene ice streams, Ann. Glaciol., 28, 67–74, doi:10.3189/172756499781821625, 1999.

Swindles, G. T., Morris, P. J., Whitney, B., Galloway, J. M., Gałka, M., Gallego-Sala, A., Macumber, A. L., Mullan, D., Smith, M. W., Amesbury, M. J., Roland, T. P., Sanei, H., Patterson, R. T., Sanderson, N., Parry, L., Charman, D. J., Lopez, O., Valderamma, E., Watson, E. J., Ivanovic, R. F., Valdes, P. J., Turner, T. E. and Lähteenoja, O.: Ecosystem state shifts during long-term development of an Amazonian peatland, Glob. Chang. Biol., doi:10.1111/gcb.13950, 2017.

Tsai, V. C., Stewart, A. L. and Thompson, A. F.: Marine ice-sheet profiles and stability under Coulomb basal conditions, J. Glaciol., 61(226), 205–215, doi:10.3189/2015JoG14J221, 2015.

Valdes, P. J., Armstrong, E., Badger, M. P. S., Bradshaw, C. D., Bragg, F., Crucifix, M., Davies-Barnard, T., Day, J. J., Farnsworth, A., Gordon, C., Hopcroft, P. O., Kennedy, A. T., Lord, N. S., Lunt, D. J., Marzocchi, A., Parry, L. M., Pope, V., Roberts, W. H. G., Stone, E. J., Tourte, G. J. L. and Williams, J. H. T.: The BRIDGE HadCM3 family of climate models: HadCM3@Bristol v1.0, Geosci. Model Dev., 10(10), 3715–3743, doi:10.5194/gmd-10-3715-2017, 2017.

van der Veen, C. J.: Fundamentals of glacier dynamics, CRC Press., 2013.

[revised manuscript text omitted]

**Figure 7: Results of the ensemble of *READVANCE* simulations testing instability in the Minch Ice Stream.** Contemporary coast is shown as a thin grey line. (a, b) Evolution of the ice sheet area (a) and volume (b) over the Minch sector in the *RETREAT* (black) and *READVANCE* simulations (coloured lines). Line colours correspond to those of Figure 6, such that the initialisation of the *READVANCE* simulations is 400 years later than the timing of the grounding line positions shown in Figure 6. Labels c and d indicate the "maximum" (c) and "collapsed" (d) stable states with corresponding panels showing the respective surface ice sheet velocity (m/y) and grounding line locations (purple line).

[Figure]

**Figure 8: The difference in the effects of MISI between the East Trough and the West Trough. a) 6,400 model year *RETREAT* margin (green) and the stable "collapsed" extent margin (purple). b) A-A' transect of bathymetry and collapse extent in the East Trough. c) B-B' transect of bathymetry and collapse extent in the West Trough.**